# Cortical response states for enhanced sensory discrimination

**Diego A Gutnisky[1,2†], Charles Beaman[1†], Sergio E Lew[1,3], Valentin Dragoi[1]\***

[1]Department of Neurobiology and Anatomy, McGovern Medical School, University of Texas, Houston, United States; [2]Janelia Research Campus, Howard Hughes Medical Institute, Ashburn, United States; [3]Instituto de Ingeniería Biomédica, Universidad de Buenos Aires, Argentina, South America

**Abstract** Brain activity during wakefulness is characterized by rapid fluctuations in neuronal responses. Whether these fluctuations play any role in modulating the accuracy of behavioral responses is poorly understood. Here, we investigated whether and how trial changes in the population response impact sensory coding in monkey V1 and perceptual performance. Although the responses of individual neurons varied widely across trials, many cells tended to covary with the local population. When population activity was in a 'low' state, neurons had lower evoked responses and correlated variability, yet higher probability to predict perceptual accuracy. The impact of firing rate fluctuations on network and perceptual accuracy was strongest 200 ms before stimulus presentation, and it greatly diminished when the number of cells used to measure the state of the population was decreased. These findings indicate that enhanced perceptual discrimination occurs when population activity is in a 'silent' response mode in which neurons increase information extraction.

DOI: https://doi.org/10.7554/eLife.29226.001

**\*For correspondence:**
Valentin.Dragoi@uth.tmc.edu

†These authors contributed equally to this work

## Introduction

The dynamics and responsiveness of individual neurons and populations in alert animals vary widely even at short time scales. Thus, even in the absence of external stimulation, the responses of cortical neurons are highly variable. For instance, in vivo intracellular recordings have demonstrated that the membrane potential can dynamically change to produce fluctuations in spontaneous and evoked neural activity (*Azouz and Gray, 2003*; *Poulet and Petersen, 2008*). Furthermore, previous investigations using diverse recording techniques – voltage-sensitive dyes (*Shoham et al., 1999*; *Slovin et al., 2002*), two-photon calcium imaging (*Frye and MacLean, 2016*), multi-electrode arrays (*Arieli et al., 1995*), EEG (*Baumgarten et al., 2015*; *Ergenoglu et al., 2004*; *Romei et al., 2008*; *van Dijk et al., 2008*), and fMRI (*Power et al., 2011*; *Power et al., 2014*; *Yeo et al., 2011*) – have shown that spontaneous and evoked cortical activity are highly structured in space and time (*Ringach, 2009*). Consistent with these results, several recent studies have demonstrated that correlated fluctuations in spontaneous and evoked activity can influence stimulus tuning and neural coding (*Poulet and Petersen, 2008*; *Ecker et al., 2014*; *Tan et al., 2014*; *McGinley et al., 2015a*; *Pachitariu et al., 2015*; *Schölvinck et al., 2015*; *Niell and Stryker, 2010*; *Renart et al., 2010*). Indeed, individual cells can rapidly shift between various states of excitability to influence stimulus processing (*Tan et al., 2014*; *McGinley et al., 2015a*; *Arieli et al., 1996*; *Fiser et al., 2004*; *Haider et al., 2007*; *Han et al., 2008*; *Tsodyks et al., 1999*), hence raising the possibility that the response state of neuronal populations could influence sensory coding and possibly perceptual accuracy. However, despite multiple lines of evidence that the dynamics of population activity during wakefulness control how external stimuli are encoded, whether rapid fluctuations in neuronal population activity impact the accuracy of perceptual responses continues to remain poorly understood.

In principle, one could reason that fluctuations in neuronal activity could be detrimental for perception. If the responses of multiple neurons are pooled to decode neural activity, rapid fluctuations in response strength could result in conflicting reports that may limit the reliability of the code and its readout by downstream neurons. Indeed, fluctuations in local population activity could make it hard to disambiguate between strong evoked responses caused by preferred stimuli and strong responses caused by a transient increase in neuronal excitability. In this scenario, similar responses could be evoked by different incoming stimuli to potentially complicate sensory decoding and detrimentally influence perceptual performance.

We addressed these issues by simultaneously recording the activity of multiple V1 neurons while monkeys performed an orientation discrimination task, and examined the relationship between the neural activity before stimulus presentation, the discrimination capability of the stimulus evoked population response, and the accuracy of perceptual responses. While the firing of individual cells was distributed along a continuous spectrum, when population activity was in a 'low' response state, neurons had lower evoked responses and correlated variability, yet a higher accuracy to predict perceptual performance. Consistent with a recurrent model of neuronal response selectivity (*Gutnisky et al., 2017*), our results indicate that perceptual discrimination performance is improved when ongoing population activity is in a 'silent' response mode, a state in which populations of sensory neurons become most discriminative.

## Results

To examine whether fluctuations in neuronal responses are functionally significant for behavior, we performed multiple-electrode recordings in primary visual cortex (V1) of behaving monkeys using custom-made electrode grids and linear arrays (*Hansen et al., 2011*). Two monkeys were trained to discriminate the orientation of circular gratings by indicating whether two successive stimuli (target and test) had the same or different orientation (*Figure 1A*). After the monkey maintained fixation for 100–200 ms, a target stimulus was flashed for 400 ms and was followed, after a delay (ranging between 250–1050 ms), by a test stimulus of random orientation (within 5° or 10° of the target), which was briefly flashed for 200–400 ms. Both stimuli fully covered the receptive fields of the neurons recorded simultaneously in each session. Overall, we recorded and analyzed a total of 263 stimulus-responsive cells (up to 13 neurons per recording session). Examples of single-unit responses and trial stability are shown in *Figure 1—figure supplements 1–2*. Our strategy was to quantify the rapid fluctuations in population activity before stimulus presentation, and then examine whether these fluctuations help identify optimal network states for signal discrimination task performance.

Given that cortical networks act in a desynchronized manner during wakefulness (*McGinley et al., 2015a*; *Greenberg et al., 2008*; *Harris and Thiele, 2011*; *Vinck et al., 2015*), we focused on the rapid fluctuations in the magnitude of neuronal responses. Across trials, we found that although the firing of individual cells was distributed along a continuous spectrum, the responses of neurons were often associated with states of low firing and states of higher firing. Examples of 'low' and 'high' population response states are shown in *Figure 1B–C* for the brief, 200 ms, interval preceding the test stimulus presentation. We measured the trial-by-trial fluctuations in the responses of each individual neuron by dividing all the trials in a session into 'low' and 'high' pre-stimulus state trials. This division into low/high responses was based on the median ongoing activity in the 200 ms interval before the test stimulus presentation (*Figure 1D* shows an example cell recorded during the behavioral task in different pre-stimulus response states; see *Figure 1—figure supplement 3* for the distribution of firing rates during the pre-test period). The choice of the 200 ms interval was based on our expectation that the response immediately preceding stimulus presentation would be more likely to influence the evoked responses (*Gutnisky et al., 2017*; *Engel et al., 2016*) and perceptual performance than the more remote pre-stimulus responses (e.g., at the beginning of the delay period, but see Figure 3G). Even though we found a continuum of response states in individual neurons across trials – pre-stimulus firing rates in each session were unimodal rather than reflecting a bimodal distribution (*Figure 1E*) – our strategy to divide the trials by the median ongoing activity enabled us to compare the network and behavioral performance between two equally-sized data sets.

For our population of cells, the median change in firing rate between the two pre-stimulus states (low and high) was 18.46 ± 1.04 spks/s (median ± sem, *Figure 1E*, inset). We also confirmed previous findings that the evoked firing rate is correlated to the pre-stimulus firing rate (*Tsodyks et al., 1999*;

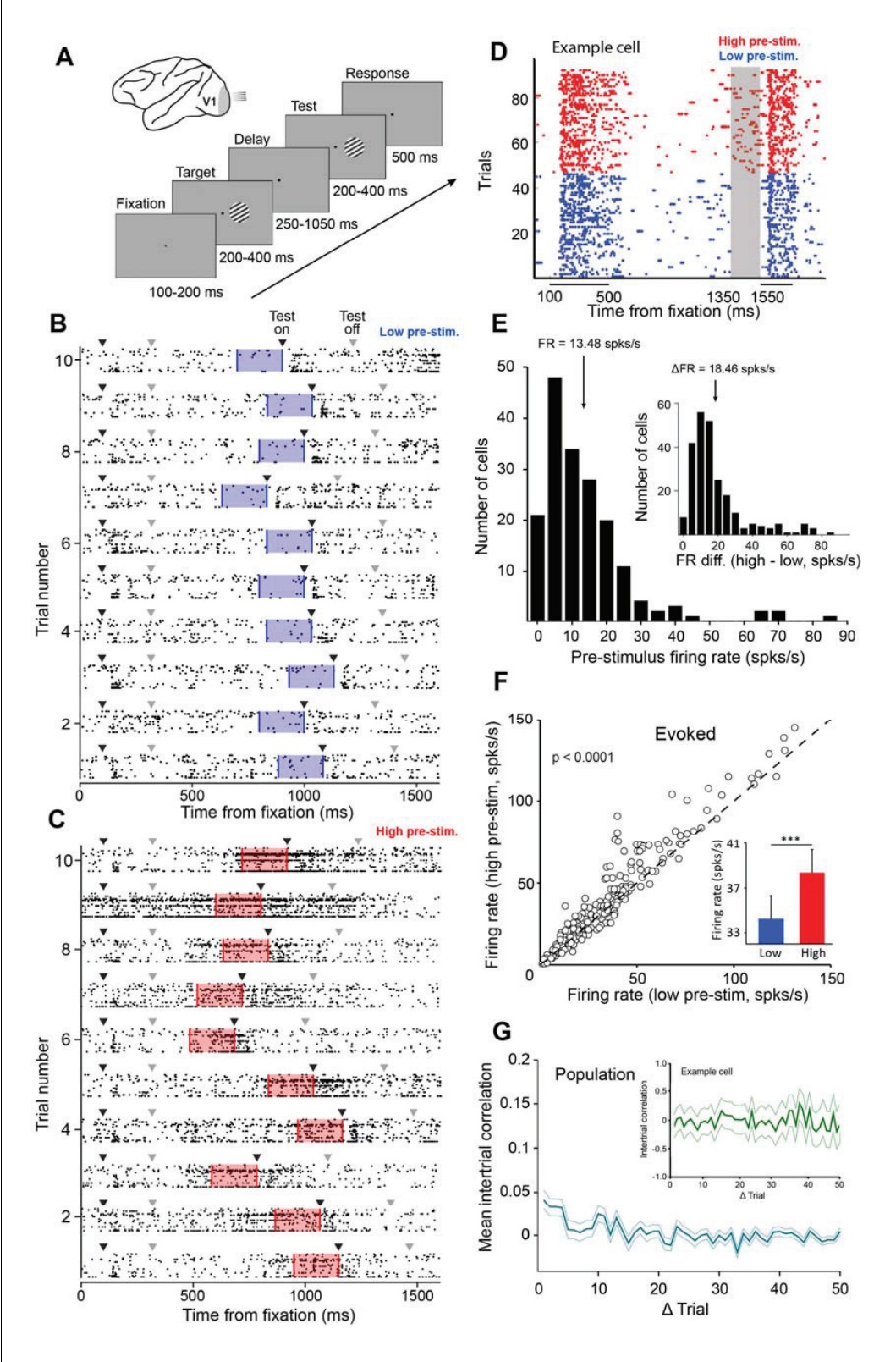

**Figure 1.** Examining the impact of cortical response state on V1 responses. (A) Schematic representation of the experimental setup. Monkeys were trained to discriminate the orientation of lines by signaling whether two successive gratings (target and test) had the same or different orientation. If the two gratings had the same orientation, the monkey was required to release the bar within 500 ms of the offset of the test stimulus. If the target and test stimuli had different orientations, the monkey was required to continue holding the bar (the number of match and non-match stimuli was identical). (B–

*Figure 1 continued on next page*

*Figure 1 continued*

C) Ten example trials of 18 simultaneously recorded neurons exhibiting low (panel B) or high (panel C) firing activity before test stimulus presentation. Black arrowheads represent stimulus onset and gray arrowheads represent stimulus offset. The blue (panel B) and red (panel C) shaded regions represent the 200 ms interval before test stimulus presentation. (D) Neuronal activity during the delayed-match-to-sample orientation discrimination task. The gray shaded area indicates the interval used to classify trials into low and high pre-stimulus state trials. The colored dots represent the timing of action potentials of an example neuron in the low (blue) and high pre-stimulus trials (red). (E) Population histogram of pre-stimulus firing rates measured 200 ms before the presentation of the test stimulus. The mean pre-stimulus rate was 13.48 ± 0.86 spks/s. (Inset) histogram of firing rate change between the two pre-stimulus states. The median change in firing rate between the two states was 18.46 ± 1.04 spks/s (arrow marks bin). (F) Evoked firing rate in the high pre-stimulus response state plotted as a function of the evoked firing rate in the low pre-stimulus state (each circle corresponds to one cell). Evoked firing rate was significantly greater in the high pre-stimulus state (p<0.0001, paired t-test). (G) Mean correlation in pre-stimulus firing rates between trials for the entire population of cells (correlation values are near chance level). Error bars represent SEM. (inset) Auto-correlation of pre-stimulus firing rates for one example cell showing no significant correlation in trial-by-trial pre-stimulus firing rates (green shadow represents the 95% confidence intervals).

DOI: https://doi.org/10.7554/eLife.29226.002

The following figure supplements are available for figure 1:

**Figure supplement 1.** Spike sorting example.
DOI: https://doi.org/10.7554/eLife.29226.003
**Figure supplement 2.** Neural stability over the recording session.
DOI: https://doi.org/10.7554/eLife.29226.004
**Figure supplement 3.** Distribution of mean firing rates in the low and high pre-stimulus periods for all the neurons recorded in the behavioral experiments.
DOI: https://doi.org/10.7554/eLife.29226.005
**Figure supplement 4.** Baseline-subtracted evoked responses as a function of pre-stimulus response state for all the neurons recorded in the behavioral experiments.
DOI: https://doi.org/10.7554/eLife.29226.006

*Gutnisky et al., 2017*; *Kohn and Smith, 2005*) – as expected, and confirming previous reports (*Gutnisky et al., 2017*; *Poort and Roelfsema, 2009*; *Arandia-Romero et al., 2016*), the evoked firing rate was significantly greater in the high pre-stimulus state (*Figure 1F*, 38.36 ± 2.17 vs. 34.24 ± 1.92 spks/s, p<0.0001, paired t-test). Subtracting pre-stimulus activity from evoked responses reveals that stimulus-driven fluctuations in neuronal responses are smaller than the fluctuations in ongoing activity (*Figure 1—figure supplement 4*). This argues that, during wakefulness, evoked responses cannot be predicted from the simple addition of the deterministic response and the preceding ongoing activity, as suggested by earlier studies in anesthetized animals (*Arieli et al., 1996*). Additionally, we examined the relationship between the response magnitude and variability of single neurons by calculating the neurons' Fano factor (*Churchland et al., 2010*; *Goris et al., 2014*; *Churchland et al., 2011*) in the low and high pre-stimulus trials. We found that normalized variability was slightly higher in the high pre-stimulus state vs. the low pre-stimulus state (1.11 ± 0.69 vs. 1.02 ± 0.41, p<0.05, paired t-test). We next determined whether the pre-stimulus response state of the neurons in our population is correlated across trials. Across sessions, we failed to find a temporal structure in the trial fluctuations of pre-stimulus firing rate. Indeed, for the population of cells the auto-correlation of the pre-stimulus firing rates across trials reveals the absence of a temporal structure (*Figure 1G*, p>0.05, Wilcoxon rank-sum, Bonferonni-corrected). This indicates that the V1 cells undergo seemingly random fluctuations in pre-stimulus activity across trials.

## Neurons tightly coupled with local population predict state-dependent changes in behavior

We first investigated whether the pre-stimulus activity of individual cells is correlated with that of the population response. Thus, we examined the relationship between the trial-by-trial response state of one individual neuron and the mean response of the rest of the recorded neurons in the population. For each cell, we computed the probability that a cell shares the same pre-stimulus state (low or high) as the remaining population (*Figure 2*). That is, we first normalized the firing rates across trials between 0 (minimum firing rate) and 1 (maximum firing rate) independently for each neuron. Then, we computed the median firing rate of that cell and categorized a pre-stimulus response in a given trial as low or high depending on whether the response was below or above the median rate. We next computed the mean normalized activity for the entire population of cells while excluding the

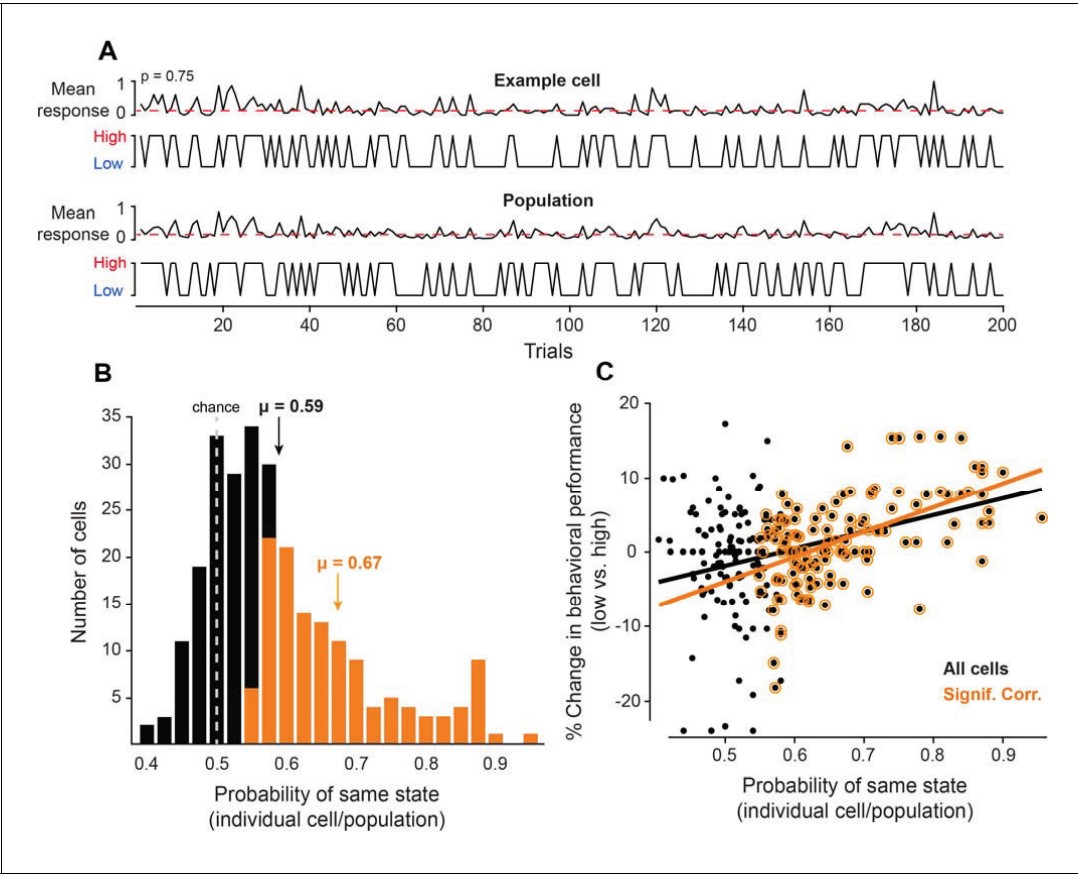

**Figure 2.** Neurons coupled with local population predict state-dependent changes in behavior. (**A**) (Top) Firing rate of one example cell normalized between 0 and 1 for 200 trials. The red dotted line represents the median normalized firing rate. Trials with firing rates below the median are placed in the low pre-stimulus firing rate group and trials above the median are placed in the high pre-stimulus category. (Bottom) The mean firing rate of the remaining simultaneously recorded population of cells (excluding the example cell) for this example session, normalized between 0 and 1. Trials with firing rates below the median are placed in the low pre-stimulus firing rate group and trial above the median are placed in the high pre-stimulus group. For the example cell in panel A, the probability of sharing the same pre-stimulus state with the remaining population is p=0.75. (**B**) Histogram of the probability of same state (individual cells-population) for the entire set of cells for all sessions. The dotted line represents the chance probability level of 0.5 for a cell to share the same pre-stimulus state as the population. The mean probability is $\mu = 0.59$, which is significantly greater than chance ($p<0.0001$, Wilcoxon rank-sum). The orange bars represent a sub-population of neurons that are significantly correlated to the population activity. The mean probability of the significantly correlated sub-population is $\mu = 0.67$. (**C**) Percent change in behavioral performance (low vs. high pre-stimulus state) as a function of the probability of same pre-stimulus state for each neuron. Positive/negative numbers: behavioral performance is improved/impaired in the 'low' pre-stimulus state. The two variables are significantly correlated for all neurons ($r = 0.36$, $p<10^{-8}$, Pearson correlation). The black line represents the linear regression fit ($R^2 = 0.13$). The change in behavioral performance and probability of same pre-stimulus state were more correlated for the sub-population of neurons that are significantly correlated to the population response ($r = 0.55$, $p<10^{-10}$, Pearson correlation). The orange line represents the linear regression fit ($R^2 = 0.30$).

DOI: https://doi.org/10.7554/eLife.29226.007

neuron of interest. As illustrated in *Figure 2A*, this analysis allowed us to track the trial-by-trial fluctuations in the responses of one cell as a function of the population response. Relating the responses of one neuron to the population response has clear advantages over other methods, such as pairwise correlations, because it relies on the response of the entire population, not just two cells, and can be computed in each trial. If the trial-by-trial response state of a cell is synchronized to the population response, the probability that that cell shares the same pre-stimulus state as the remaining population is 1. If, on the other hand, the response state of a given cell is independent of the population response, the probability that that cell shares the same response state as the remaining population would be significantly lower.

Based on the trial fluctuations in the responses of individual neurons and the rest of recorded cells, we calculated the probability that a given cell is in the same pre-stimulus state as the

population (*Figure 2B*). Overall, the mean probability of the 'same' response state for our group of neurons was μ = 0.59 ± 0.04, which was significantly greater than chance level (*Figure 2B*, p<0.0001, Wilcoxon rank-sum), thus demonstrating that the pre-stimulus state of individual neurons is correlated with that of the remaining population. However, whereas some cells were highly correlated with the population, others were less correlated (*Figure 2B*), in agreement with previous experimental findings measuring the functional coupling between individual neurons and their local network (*Okun et al., 2015*). Across sessions, we found that 130 out of 263 cells were significantly correlated to the population activity (p<0.05, Binomial distribution); the mean probability of the significantly correlated subpopulation is μ = 0.67 ± 0.01. Can individual neurons predict the fluctuations in behavioral performance during the task? We reasoned that those cells that are tightly coupled with the local population response are more likely to predict the fluctuations in perceptual accuracy than the cells with a low coupling probability.

To address this issue, we examined the relationship between the behavioral performance corresponding to the low vs. high pre-stimulus state of a given cell and the probability that that cell shares the same response state as the population. Specifically, for each cell we divided the trials into two groups, low and high pre-stimulus, based on whether the pre-stimulus firing rate was below or above the median pre-stimulus firing rate across trials. We then analyzed the animal's behavioral performance in the session when the test orientation was within 10° of the target (by pooling the positive and negative orientation differences), separately for the low and high state trials defined by the responses of only one neuron. For this analysis and the subsequent ones, we did not include the 'easier' discrimination trials in which the test stimulus was > 10° away from the target because that condition was associated with a much smaller number of incorrect responses, that is, behavioral performance was close to the ceiling level.

The trial analysis allowed us to calculate the correlation between the % change in discrimination performance (low vs. high) and the probability of each cell being in the same state as the population. The results confirmed our expectation that cells that are tightly coupled with the population are more likely to predict the changes in perceptual accuracy when the pre-stimulus firing rate changes from low to high state. Indeed, as shown in *Figure 2C*, there was a significant relationship between behavioral performance and the degree of coupling between individual neurons and the population (all cells: r = 0.36, p<0.0001, Pearson correlation; significant cells: r = 0.55, p<$10^{-10}$, Pearson correlation). Importantly, since the response fluctuations of individual cells were significantly correlated with the population response, these results raise the possibility that by pooling neurons we could increase our estimate of the true cortical response state and ultimately improve the predictability of behavioral performance when the population firing rate changes from the low to high state.

## Pooling neurons improves the predictability of behavioral responses

We further explored the impact of fluctuations in population activity on behavioral performance by pooling the cells recorded in the same session. We performed this analysis by gradually increasing the size of the neuronal pool. That is, for each session we analyzed all possible combinations of neuronal subpopulations of size 1 to *n* (*n* is the number of simultaneously recorded cells within a session). For instance (*Figure 3A*), for a population of three cells – A, B, and C – there are three possible sets of individual cells (A, B, and C), three possible sets of two cells (i.e., AB, AC, BC), and one set of three cells (ABC). For each pool of neurons, we calculated the mean normalized pre-stimulus activity across all the cells within each pool (independently for different population sizes) and divided the trials into low and high pre-stimulus activity trials, as described previously. Subsequently, we compared the behavioral performance for low and high pre-stimulus states as a function of population size. First, we expected to confirm our prediction that monkey's orientation discrimination performance is better in the low pre-stimulus state. Second, we expected to find that increasing the population size (in order to obtain better estimates of network's state) yields a larger difference in behavioral performance between the low and high pre-stimulus states.

Motivated by the results in *Figure 2C*, we first examined whether behavioral performance in the orientation discrimination task is improved in the low pre-stimulus state. Thus, we analyzed behavioral performance when the relative difference between the target and test (Δθ) was ±5° (pooling the positive and negative orientation differences) in n = 11 sessions. We found (*Figure 3B–C*) that perceptual accuracy was highest when neurons were in the low pre-stimulus response state (p<0.0001, Wilcoxon rank-sum at highest population level; F(1,103)=23.3; p<0.0001; two-way repeated

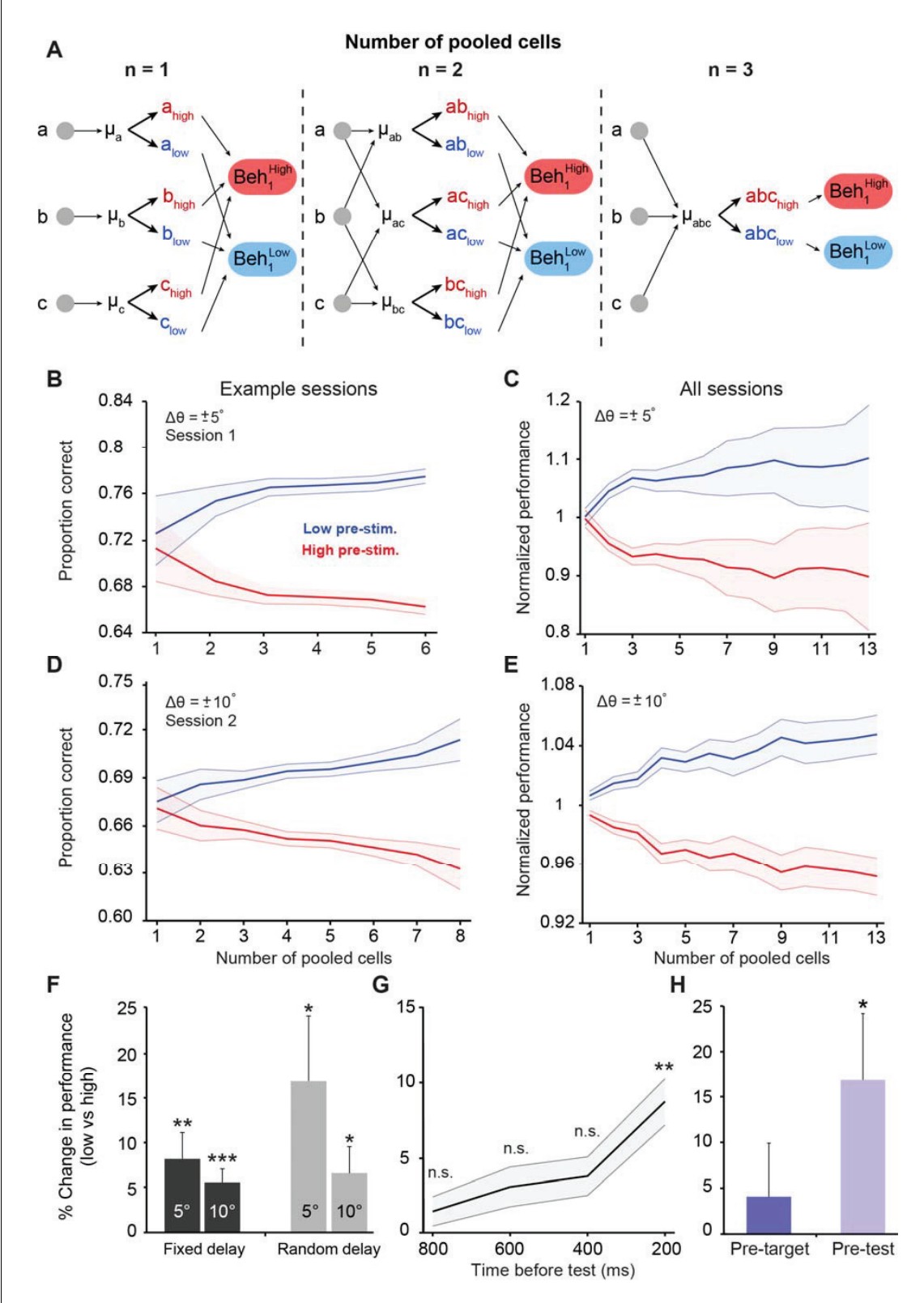

**Figure 3.** Cortical state influences behavioral performance in an orientation discrimination task. (A) Diagram depicting our analysis linking neuronal populations and behavior (examples provided for n = 1, 2, and 3 cells). For n = 1, we computed the mean pre-stimulus firing rate individually for each neuron ($\mu_A$, $\mu_B$, and $\mu_C$) and then calculated the average behavioral performance in the low and high pre-stimulus trials, averaged across the three cells for each group (Beh$_1^{low}$ and Beh$_1^{high}$). For n = 2, we computed the mean normalized pre-stimulus activity for each pool of 2 cells ($\mu_{AB}$, $\mu_{AC}$, and $\mu_{BC}$),
*Figure 3 continued on next page*

*Figure 3 continued*

then divided the trials into high and low groups, and calculated the average behavioral performance in the low and high pre-stimulus trials ($Beh_2^{low}$ and $Beh_2^{high}$). For n = 3, we computed the mean normalized response for all three cells ($\mu_{ABC}$) and then split the trials to compare behavioral performance between the low and high pre-stimulus groups ($Beh_3^{low}$ and $Beh_3^{high}$). (B–E) Behavioral performance is modulated by the ongoing population activity; a single session (panels B and D); all sessions (panels C and E). Behavioral performance associated with each ongoing activity state in each session was normalized by dividing the performance in each state by the average session performance (irrespective of pre-stimulus state). The pre-stimulus state was determined based on the pooled activity of neural populations of varying size (based on the method in panel A). The difference in discrimination performance between low and high pre-stimulus response states was greater when population size increases. Panels 3B and C correspond to orientation differences between target and test of ±5°; Panels 3D and E correspond to orientation differences between target and test of ±10°. 'All sessions' in panel C includes data from monkeys 1 and 2 (n = 24 sessions). 'All sessions' in panel E includes data from monkeys 1, 2, and 3 (n = 42 sessions). Error bars represent s.e.m of session performance for each population size. (F) Behavioral improvement in the low vs. high pre-stimulus conditions is present for both the fixed and random delay conditions for the ±5° and ±10° orientation differences (*p<0.05, **p<0.01, ***p<0.001; Wilcoxon signed-rank test). Results in panels F-H were obtained for a population size of 5 to include most sessions in the analysis. (G) Behavioral improvement in the low vs. high pre-stimulus state for different pre-stimulus periods. The x-axis represents the time relative to the onset of the test stimulus. Pre-stimulus state was assessed based on the pre-stimulus interval in 200 ms steps (during the delay period). The improvement in discrimination performance in the low pre-stimulus state occurs only during the 200 ms period before test presentation. (**p<0.01; n.s. = non significant; Wilcoxon signed-rank test). Error bars represent s.e.m. (H) Neuronal activity in the pre-test interval influences behavior to a larger extent than that in the pre-target interval (*p<0.05; n = 13 sessions from the random delay experiment; the fixed delay data was not included because the pre-target interval was too small, that is, 100 ms).

DOI: https://doi.org/10.7554/eLife.29226.008

The following figure supplements are available for figure 3:

**Figure supplement 1.** Cortical state influences behavioral performance in an orientation discrimination task (fixed delay experiments).
DOI: https://doi.org/10.7554/eLife.29226.009

**Figure supplement 2.** Cortical state influences behavioral performance in an orientation discrimination task (results shown for each monkey trained in the fixed delay experiments).
DOI: https://doi.org/10.7554/eLife.29226.010

**Figure supplement 3.** Behavioral performance depends on task difficulty.
DOI: https://doi.org/10.7554/eLife.29226.011

**Figure supplement 4.** Cortical state influences behavioral performance in an orientation discrimination task (results shown for the random delay experiments).
DOI: https://doi.org/10.7554/eLife.29226.012

**Figure supplement 5.** Pre-target neural activity does not influence behavioral performance in a significant manner.
DOI: https://doi.org/10.7554/eLife.29226.013

**Figure supplement 6.** Relationship between LFP power and pre-stimulus response state.
DOI: https://doi.org/10.7554/eLife.29226.014

measures ANOVA). Similar results were observed when Δθ was ±10° (n = 29 sessions, *Figure 3D–E*; p<0.0001, Wilcoxon rank-sum at highest population level; F(1,425)=104.4; p<0.0001, two-way repeated measures ANOVA; see also *Figure 3—figure supplements 1–2* for individual animal performance). Importantly, the difference in discrimination performance between the low and high pre-stimulus activity conditions was amplified when the number of cells used to measure the network state was increased (r = 0.84, p<0.0001; Pearson correlation for ±5° and r = 0.93, p<0.0001; Pearson correlation for ±10°), possibly due to a better estimation of the 'true' cortical state in larger pools of neurons. In addition to pre-stimulus response state, behavioral performance also depends on task difficulty. Indeed, animals performed more poorly (*Figure 3—figure supplement 3*) when they discriminated small orientation differences (±5°; mean across sessions: 64.58 ± 2.19% correct responses) compared to less difficult task conditions (±10°; mean across sessions: 79.17 ± 1.85% correct responses, p=4.03 $10^{-5}$, rank-sum test). The impact of pre-stimulus response state on behavioral performance depended on task difficulty – as shown in *Figure 3F*, the change in behavioral performance (low vs. high pre-stimulus state) was significantly larger for the more difficult orientation difference (p<0.01, Wilcoxon signed-rank; to maximize the number of sessions, the results for the 'n = 5' population size are shown).

Next, we investigated whether the modulation of behavioral performance by pre-stimulus response state could be explained by the 'top-down' anticipation of the test stimulus. To address this issue we took advantage of our experimental design in which some of the sessions had a fixed delay while others had a random delay. Indeed, in a subset of our experiments, the delay between

the target and test stimuli was fixed at 1050 ms (11 out of 24 sessions for Δθ ± 5°; 29 out of 42 sessions for Δθ ± 10°), possibly allowing the animal to predict the appearance of the test stimulus. Although stimulus expectation in itself cannot invalidate our results, we nonetheless compared our fixed delay experimental procedure to a randomized target-test delay interval (which could vary between 250 and 750 ms, *Figure 3—figure supplement 4*). In the randomized delay condition, we isolated 131 visually responsive neurons in one animal across 13 sessions. However, even when the monkey was less able to predict the time of test stimulus presentation, the behavioral performance modulation by pre-stimulus activity was relatively similar to that observed in our original fixed-delay experiments (*Figure 3F*) –performance was improved in the low pre-stimulus condition for both the ±5° (p<0.05, Wilcoxon rank-sum at highest population level; F(1,259)=42.14, p<0.0001; two-way repeated measures ANOVA) and ±10° discriminations (p<0.005, Wilcoxon rank-sum at highest population level; F(1,251)=55.7; p<0.0001, two-way repeated measures ANOVA, *Figure 3—figure supplement 4B–D*). For comparison, when delay was fixed behavioral performance was significantly improved in the low pre-stimulus state when Δθ was ±5° (p<0.0001, Wilcoxon rank-sum at highest population level; F(1,355)=56.1, p<0.0001, two-way repeated measures ANOVA) (*Figure 3B–C*), and similar results were observed when Δθ was ±10° (p<0.0001, Wilcoxon rank-sum at largest population level; F(1,697)=164.6, two-way repeated measures ANOVA, p<0.0001) (*Figure 3D–E*). Altogether, these results indicate that the modulation of behavioral performance by the population pre-stimulus activity cannot be entirely explained by top-down effects such as expectation or attention (*McAdams and Maunsell, 1999*; *Luck et al., 1997*; *Yoshor et al., 2007*).

We further examined whether the difference in behavioral performance between the two pre-stimulus response states (low and high) could also be predicted from the more remote neuronal responses, not only the activity immediately preceding the stimulus. We thus selected successive 200-ms pre-stimulus activity windows for each population of cells starting 200, 400, 600, and 800 ms prior to stimulus presentation, and repeated our original analysis. However, we found that only the pre-stimulus activity immediately preceding the test presentation was able to predict behavioral performance (*Figure 3G*) – analyzing the pre-stimulus in beyond 200 ms did not provide information about behavioral outcomes (p>0.05). Thus, the impact of pre-stimulus response state on behavioral decisions is based on the ongoing activity immediately preceding the test stimulus.

Does prediction of behavioral performance only occur in relation to the presentation of the stimulus closest in time to the behavioral response (i.e., the test) or is common to both the target and test stimuli? In principle, one could argue that behavioral decisions in the discrimination task are likely made based on comparing the neuronal responses elicited by the target and test stimuli. Therefore, we reasoned that analyzing the pre-stimulus activity preceding the target will likely demonstrate a similar dependence of perceptual accuracy on the pre-target ongoing activity. We examined this issue by conducting a complementary analysis of behavioral performance in relation to neuronal firing 200 ms before *target* stimulus presentation by analyzing the random-delay data set (which included a longer, 300 ms, pre-target fixation window). However, while there was a tendency for behavioral performance to increase in the low firing pre-target interval, performance was not significantly modulated by the pre-target response state (p>0.1, *Figure 3H*). We further repeated the analyses in *Figure 3B–E* and found (*Figure 3—figure supplement 5*) that behavioral performance was independent of the pre-target response state: when Δθ was ±5°: p=0.40, Wilcoxon rank-sum; when Δθ was ±10°: p=0.09, Wilcoxon rank-sum (we used the largest population size for both comparisons). Taken together, these results indicate that only the fluctuations in the population response before the presentation of stimuli that are closest in time to behavioral responses will influence behavioral performance.

Finally, we examined whether the fluctuations in cortical response state are correlated with global arousal or they reflect local changes in network activity. Therefore, we performed additional analyses in which we examined the correlation between pre-stimulus response to the test stimulus and pupil size, which is a measure of global arousal (*Scharinger et al., 2015*; *Joshi et al., 2016*). That is, pupil dilation is associated with increased arousal, whereas pupil constriction is associated with diminished arousal. In a subset of sessions (n = 11) examining the 200 ms pre-stimulus test response revealed that out of 131 cells, only 12 exhibited significant trial-by-trial correlation with mean pupil size (p<0.05, Pearson correlation; of these 12 cells, 7 showed a positive correlation between the pre-stimulus response and pupil size). However, when we calculated the population response, none of the sessions exhibited a significant trial correlation between the population pre-stimulus response

and pupil size (p>0.05, Pearson correlation). Additionally, when we divided trials into two groups based on median pupil size (low vs. high pupil size), there was no significant difference between the mean pre-stimulus firing rates in the two groups (p>0.1, Wilcoxon signed rank test, pooling all the cells in our sample). This analysis demonstrates that global fluctuations in brain state, measured by pupil size, do not seem to explain the trial-by-trial fluctuations in pre-stimulus response state discussed here.

## Cortical state influences encoded information

The state-dependent changes in behavioral performance shown in *Figure 3* raise the possibility that trial fluctuations in pre-stimulus population activity may influence the accuracy with which V1 cells can extract stimulus orientation. Thus, we examined the amount of information encoded in population activity by using the Fisher linear discriminant (FLD) and population *d'* (*Poort and Roelfsema, 2009*; *Averbeck and Lee, 2006*). Our goal was to understand why neuronal populations exhibit low evoked firing rates but nonetheless transmit more information about stimulus orientation in the low pre-stimulus state. First, to assess the ability of our recorded network of cells to discriminate orientation, we performed an optimal linear classification by computing the Fisher linear discriminant (FLD). This allows us to find the optimal multi-dimensional projection of the data (number of dimensions equal to number of simultaneously recorded neuron in a session) into one-dimensional space (*Poort and Roelfsema, 2009*; *Averbeck and Lee, 2006*) that best separates the two different stimuli (target and test orientations), a task which the brain must accomplish in order to receive reward. The FLD projection maximizes the distance between the means of two groups of data while minimizing the variance within each group (*Singer and Kreiman, 2011*). Thus, the task of the FLD neural decoder is to discriminate between the two stimuli, exactly as animals were required to do in the task. Better decoder performance corresponds to an increase in sensory information (*Moreno-Bote et al., 2014*).

For each session, we analyzed the capacity of the classifier to separate between the target and test stimuli when the pre-stimulus population response is low or high (see Materials and methods). For instance, *Figure 4* represents 13 simultaneously recorded neurons from one example session (*Figure 4A*). When only two cells are used to linearly separate the two stimuli (*Figure 4B* shows the number of spikes of one pair of cells, neurons 3 and 9, in all the trials for both the target and test stimuli), the classifier performs somewhat better in the low pre-stimulus response state. The axes of the histograms are plotted perpendicular to the FLD dimension that maximizes the separability between the two stimuli (green lines), and the solid curves represent one-dimensional Gaussian fits for the target and test response distributions. There is a better linear separation between the response distributions corresponding to the target and test stimuli in the low pre-stimulus response state when only two cells are used. However, when the FLD analysis is applied to the entire population of 13 cells, there is a pronounced increase in the separability between the target and test stimuli in the low pre-stimulus state (*Figure 4C*), consistent with the changes in behavioral performance.

Population discriminability between two multivariate distributions using the FLD method can be quantified by a single variable $d^2 = \left(\bar{x}_A - \bar{x}_B\right)^T Q^{-1} \left(\bar{x}_A - \bar{x}_B\right)$ (*Averbeck and Lee, 2006*), where $\bar{x}_A$ and $\bar{x}_B$ represent the vector of mean firing rates in all trials for target and test, respectively, and $Q^{-1}$ represents the inverse of the pooled covariance matrix. The probability of correct classification (PCC) is directly related to $d^2$ by the complementary error function (see Materials and methods for details). We thus computed the PCC for each session (*Figure 5A*) after measuring cortical state (low or high pre-stimulus response state) based on the pre-stimulus activity of a variable number of cells in each session (from 1 to 13). Importantly, the performance of the decoder was measured using all the neurons in the population. For instance, decoder performance corresponding to n = 1 represents the mean population decoder performance (considering all the cells in the population) for which the low and high pre-stimulus states were measured based on the pre-stimulus response of only one neuron (by averaging the decoder performances after diving trials into low vs. high pre-stimulus state based on each neuron in the population). Since, on average, the cortical state of one cell is coupled to local population activity (*Figure 2*), decoder performance is relatively high (around 0.82) even when only one neuron is used to measure pre-stimulus state. As expected, PCC increases as the number of cells that were used to distinguish between low and high response states increases from 1 to 13.

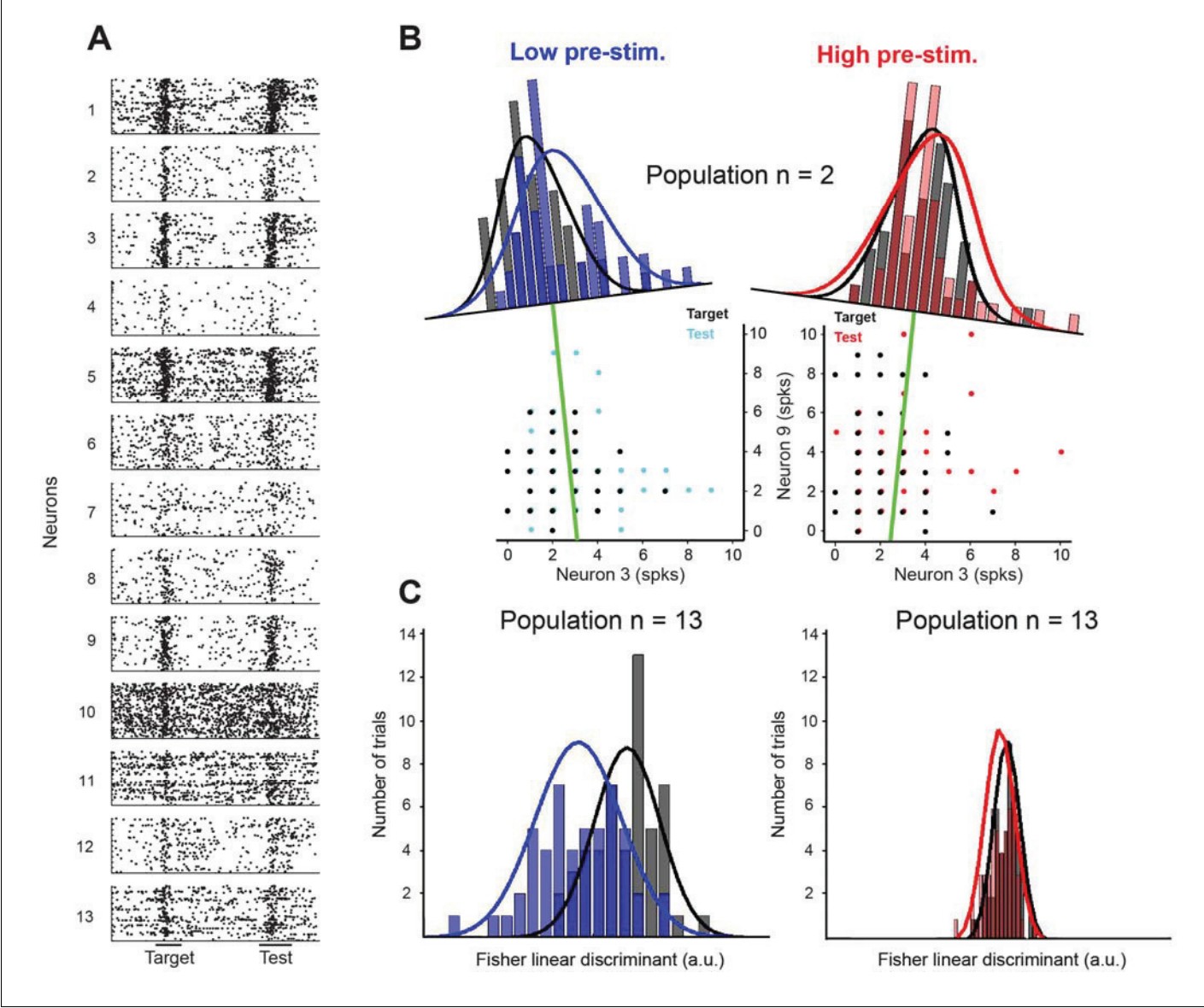

**Figure 4.** Fisher linear discriminant (FLD) analysis for one example session. (A) Raster plots for 13 cells recorded simultaneously in a session. Each dot represents the time of an action potential. Horizontal bars at the bottom represent stimulus duration for target and test. The random delay period has been truncated to align the test responses. (B) Example FLD of one cell pair (neurons 3 and 9). Each circle represents the total number of spikes elicited during the target or test stimulus. Each histogram is plotted on the Fisher linear discriminant axis which maximizes the difference between target and test relative to the variance of the responses. The black and blue (black and red for the high pre-stimulus condition) curves represent one-dimensional Gaussian fits for the target and test distributions, respectively. The green line represents the decision boundary. (C) Example FLD for the full population of 13 cells. There is a greater difference between the two distributions in the low pre-stimulus trials compared to the high pre-stimulus trials.

DOI: https://doi.org/10.7554/eLife.29226.015

By averaging decoder performance across sessions and animals, we found that PCC was significantly greater in the low pre-stimulus response trials ($p < 0.05$; paired t-test computed for the highest population level for each session). We also compared the difference between decoder performance (low vs. high pre-stimulus response states) at small ($N = 2$) and large ($N = 12$) population sizes and found that the performance difference was greater when the population size is large ($p < 0.05$, bootstrap test; *Figure 5A* inset). The difference between decoder performances in the low vs. high pre-stimulus states was abolished when pre-stimulus activity was subtracted from the evoked responses (*Figure 5—figure supplement 1*), although, as expected, PCC increases when the number of cells

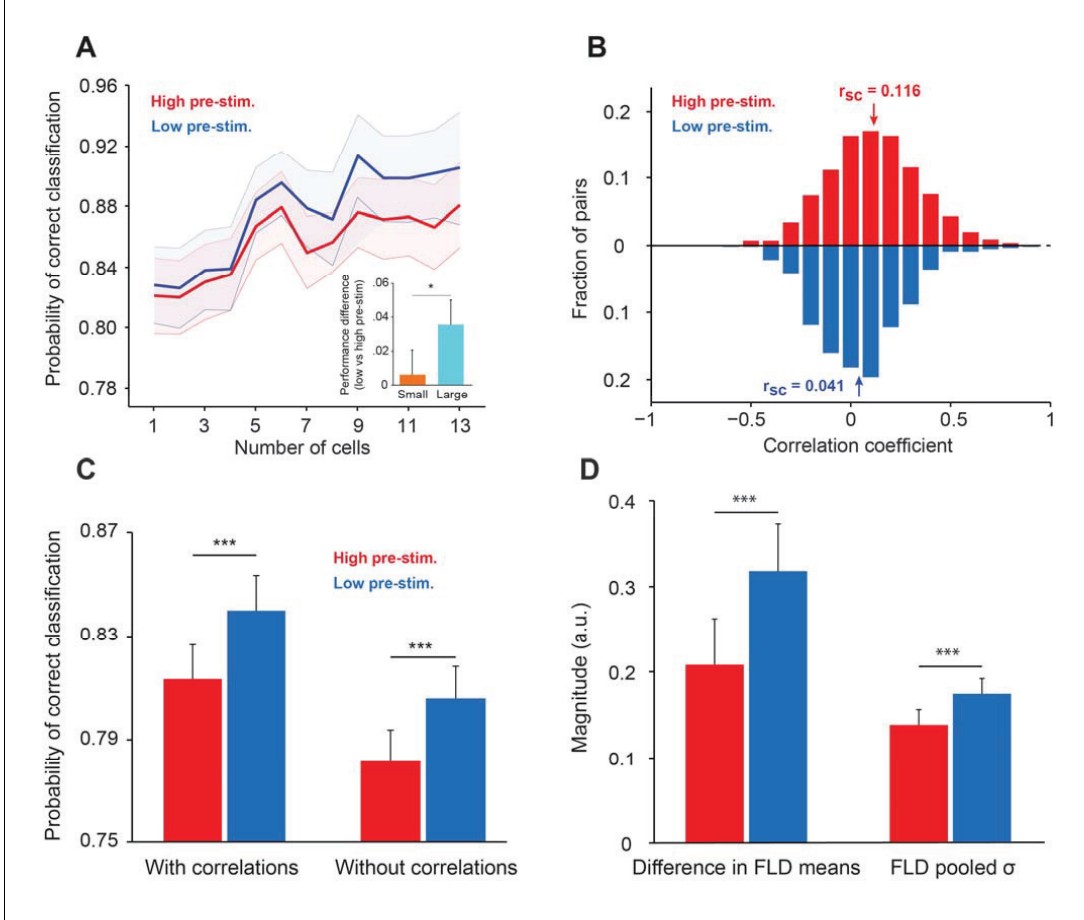

**Figure 5.** Cortical state influences the information encoded in population activity. (**A**) The probability of correct classification (PCC) as a function of population size. PCC was significantly higher in the low pre-stimulus case (F(1,164)=9.32; p<0.005; two-way repeated measures ANOVA). (inset) The difference in classification performance between low and high pre-stimulus response states for a *small* (n = 2, orange) and a *large* population (n = 12, blue). The performance difference was greater for the larger neural population (p<0.05, bootstrap test). (**B**) Noise correlations of evoked responses for the high and low pre-stimulus states – correlations were significantly higher in the high pre-stimulus state (p=7.918e-10; paired t-test). (**C**) Probability of correct classification for the two cortical states. 'With correlations' represents data using the following equation: $d^2 = \Delta\mu^T Q^{-1} \Delta\mu$; Probability of correct classification = $erfc\left(-\sqrt{d^2}\right)/2$. 'Without correlations' represents the probability of correct classification when ignoring the effect of noise correlations using ($d^2_{shuffled} = \Delta\mu^T Q_d^{-1} \Delta\mu$), where $Q_d$ is the diagonal covariance matrix. In each condition there is a statistically significant difference between the high and low pre-stimulus conditions (*p<0.05; paired t-test). (**D**) The magnitude of the difference in FLD means (left) and the magnitude of the pooled standard deviation ($\sigma$) of the FLD (right). In the low pre-stimulus condition, the difference in means was significantly greater (p<0.0001; paired t-test). The average variance was also higher in the low pre-stimulus condition (p<0.0001; paired t-test), but this had less overall impact on the population d'. The results in this figure were obtained for all the cells recorded across sessions for an orientation difference of ±5°.

DOI: https://doi.org/10.7554/eLife.29226.016

The following figure supplement is available for figure 5:

**Figure supplement 1.** Probability of correct classification (PCC) as a function of population size when baseline-subtracted evoked responses to the target and test stimuli are used.
DOI: https://doi.org/10.7554/eLife.29226.017

used to measure cortical state grows. This result indicates that removing the influence of ongoing activity on evoked responses renders the decoder of population activity insensitive to changes in pre-stimulus response state.

There are three possible reasons for the improved network discriminability in the low pre-stimulus state: the change in correlated variability, the difference in mean responses in the FLD dimension, or the decreased variance of responses in the FLD dimension (*Averbeck and Lee, 2006*). Indeed, it has long been reported that correlated firing between neurons can serve to constrain the schemes by

which the cortex encodes and decodes sensory information (*Ahissar et al., 1992*; *Cohen and Newsome, 2008*; *Ecker et al., 2010*; *Gutnisky and Dragoi, 2008*). Since the information about stimulus orientation that neurons extract depends on the neurons' response magnitude and correlated variability (*Smith and Kohn, 2008*), we analyzed whether noise correlations depend on the state of the network (low vs. high pre-stimulus response). We expected that correlations, which typically depend on mean spike counts, would be larger in the high pre-stimulus response state due to the higher evoked firing rates (*de la Rocha et al., 2007*) (*Figure 1D*). Our analysis confirmed this expectation – we found a significant difference in correlations between the two response states, i.e., correlated variability depends on the state of local population ($r_{sc}$ in the high vs. low pre-stimulus: $0.116 \pm 0.013$ vs. $0.041 \pm 0.011$; *Figure 5B*, $p = 7.918 \cdot 10^{-10}$, paired t-test).

We next estimated the effect of correlated trial-by-trial responses on information encoding, and hence computed the amount of network information when cells were uncorrelated, $d^2_{shuffled}$. Using this measure, we found that information in the uncorrelated networks was decreased for both the low and high pre-stimulus states. This result is not entirely surprising – although noise correlations are generally believed to be detrimental for coding, this is typically true when correlations are high and positive. However, there are significant negative correlations in our population of cells (30% of our correlation coefficients are negative, *Figure 5B*), and removing negative correlations has been shown to decrease the amount of information encoded in population activity (*Chelaru and Dragoi, 2016*; *Rosenbaum et al., 2017*). Furthermore, correlations have been shown to increase or decrease the amount of encoded information depending on the structure of correlations across a population of cells, not the absolute magnitude of correlation coefficients (*Moreno-Bote et al., 2014*). However, despite the fact that removing correlations decreases encoded information, the *difference* in the probability of classification between the two response states remained relatively unchanged (*Figure 5C*, $p > 0.05$; paired t-test), although the difference in classification performance between the low and high pre-stimulus states remained significant (*Figure 5C*).

Lastly, population *d'* can be deconstructed into the difference in mean responses in the most discriminant dimension (FLD) divided by the square root of the average variance in the same dimension (*Averbeck and Lee, 2006*). Either an increased difference in mean response (numerator of *d'*) or a decreased pooled variance (denominator of *d'*), or some combination of the two, can explain the improved classification performance in the low pre-stimulus state. In other words, either a larger difference in the mean responses between target and test responses or a decreased variance of responses in the low pre-stimulus condition explain the improvement in network performance. By comparing each variable, we found that classification performance was better in the low pre-stimulus condition due to a greater difference in FLD mean responses ($p < 0.0001$; paired t-test; *Figure 5D*, left). The FLD pooled standard deviation was larger in the low pre-stimulus condition, but ultimately this difference had a smaller total impact on discrimination performance than the difference in FLD means (*Figure 5D*, right). Altogether, these results are consistent with theoretical studies (*Chance et al., 2002*) and previous work in awake animals demonstrating an increase in response gain and sensitivity in the low pre-stimulus state (*Gutnisky et al., 2017*).

## Discussion

We have demonstrated that fluctuations in the response state of cortical networks during wakefulness, defined by the pre-stimulus population firing rate, can impact neuronal and behavioral discrimination performance. Our results indicate that the likelihood of correctly discriminating or recognizing incoming stimuli increases when cortical networks are in an appropriate state of excitability, possibly representing a measure of network 'preparedness' to facilitate sensory processing. Thus, perceptual discrimination performance is improved when the population activity is in a 'silent' mode in which the encoding of visual information is enhanced. The functional impact of cortical state has been previously examined in anesthetized animals (*Ecker et al., 2014*; *Schölvinck et al., 2015*; *Renart et al., 2010*), and using measures of arousal, such as pupil diameter, in awake mice (*McGinley et al., 2015a*; *Vinck et al., 2015*). More recently, it has been shown that fluctuations in V1 ongoing activity (*Gutnisky et al., 2017*) and evoked responses (*Arandia-Romero et al., 2016*) influence sensory encoding in individual neurons and populations. However, whether fluctuations in cortical population activity are able to influence the accuracy of sensory discriminations while the animal is performing a task has been unclear.

We found that decoding the population response before stimulus presentation can be used to predict subsequent neuronal performance and that a significant proportion of behavioral variance can be attributed to the internal response state of the local circuit before stimulus presentation. Importantly, the prediction of the behavioral response was not altered when the animal was unable to estimate the timing of occurrence of the test stimulus. Thus, top-down stimulus expectation might not significantly alter local ongoing activity to impact behavioral performance. Furthermore, we found that cortical state is correlated among neurons, and that those cells that covary with the population are more predictive of behavioral responses than the neurons that fire independently. Indeed, it has recently been proposed that neighboring neurons differ in their coupling to the mean population response, ranging from strongly coupled 'choristers' to weakly coupled 'soloists' (*Okun et al., 2015*). However, although population coupling has been found to be independent of the neurons' stimulus preference, we found here that neurons that are strongly coupled to the population response are more likely to predict the animal's behavioral choice based on the pre-stimulus response state.

Previous work in V1 (*Arieli et al., 1996*; *Tsodyks et al., 1999*; *Gutnisky et al., 2017*; *Arandia-Romero et al., 2016*), mainly performed by recording single neurons, has demonstrated that the pre-stimulus ongoing activity influences the strength and selectivity of evoked responses. However, these studies did not investigate whether the ongoing population activity contains information about the animal's behavioral choice. Furthermore, several recent studies have addressed the issue of how cortical state influences sensory encoding in the context of global state fluctuations. Thus, cortical state has been defined based on the level of synchrony across neurons or the LFP power (*Poulet and Petersen, 2008*; *McGinley et al., 2015a*; *Pachitariu et al., 2015*; *Schölvinck et al., 2015*; *McGinley et al., 2015b*; *Reimer et al., 2014*; *Vyazovskiy et al., 2011*). In these studies, it has been reported that cortical responses are weaker, but more correlated in the synchronized state (typically found during anesthesia and sleep) than during the desynchronized state (characterizing the awake state). However, one important issue that has long been ignored is how the within-state trial-by-trial fluctuations in the strength of population activity, extensively documented in the literature (*Arieli et al., 1996*; *Tsodyks et al., 1999*; *Gutnisky et al., 2017*; *Arandia-Romero et al., 2016*), influences the information encoded in population activity and the accuracy of perceptual reports. In our study, the fluctuations in pre-stimulus response state do not correspond to the classical definition of cortical state based on the degree of population synchrony (Beaman et al., 2017). Indeed, we found that when going from the low to high pre-stimulus state, the evoked responses and correlated variability increase. In contrast, population synchrony decreases evoked responses while increasing correlations between neurons. These differences may be due to the fact that we have measured within-state fluctuations (*Gutnisky et al., 2017*; *Arandia-Romero et al., 2016*) rather than across-state fluctuations. Importantly, our analysis shows that the distribution of pre-stimulus activity is unimodal (*Figure 1C*), suggesting a single state that is unlikely to be affected by the level of population synchrony.

Recent work in mouse V1 has explored the impact of global fluctuations in cortical state on neuronal responses and behavior during wakefulness (*Vinck et al., 2015*; *McGinley et al., 2015b*). However, there are important differences between our study and earlier investigations. In addition to the difference in species (monkey vs. mouse), our definition of cortical state is restricted to the local population activity that is monitored while the animal performs the task. In contrast, previous investigations measured global fluctuations in behavioral state, such as arousal, and their impact of neuronal responses. For instance, it was reported that rapid variations in locomotion and arousal (as measured by pupil diameter) control sensory evoked responses and spontaneous activity of individual neurons (*Vinck et al., 2015*; *Erisken et al., 2014*), and that noise correlations were lower during locomotion than quiescence, while evoked responses were stronger (*Vinck et al., 2015*; *Erisken et al., 2014*). Global state fluctuations, such as arousal, were also found to modulate the membrane potentials of auditory cortical neurons in mice trained on a tone-in-noise detection task (*McGinley et al., 2015a*). Importantly, arousal level was found to modulate behavioral performance by enhancing sensory-evoked cortical responses and reducing background synaptic activity. However, previous studies did not investigate how the fluctuations in local cortical state jointly influence the information encoded by ensembles of neurons and behavioral performance.

Our results are consistent with previous studies in other sensory systems, such as primary somatosensory cortex (S1), reporting a relationship between cortical response state and the sensitivity of

individual neurons. Specifically, it has been shown (*Fanselow and Nicolelis, 1999*) that behavioral state (quiet immobility or whisker movement) renders the somatosensory system optimally tuned to detect the presence of single vs. rapid sequences of tactile stimuli. It has also been shown (*Sachdev et al., 2004*) that postsynaptic potentials in rat S1 are influenced by 'up' and 'down' states, although, contrary to our findings, evoked responses were stronger in the hyperpolarized state. Importantly, none of these studies have analyzed population activity to relate it to the amount of encoded information and behavioral performance. In addition, our study has examined for the first time the coupling between individual neurons and population activity and its relationship with behavior (*Figure 2*).

One could possibly link the trial-by-trial fluctuations in cortical responses examined here to trial fluctuations in attention. Indeed, attention has been shown to modulate neuronal responses and sensitivity (*McAdams and Maunsell, 1999*; *Thiele, 2009*; *Cohen and Maunsell, 2009a*; *Gutnisky et al., 2009*; *Herrero et al., 2013*; *Treue and Maunsell, 1996*) as well as response variability and pairwise correlations (*Cohen and Maunsell, 2009b*; *Mitchell et al., 2009*) in early and mid-level visual cortical areas. However, our results are inconsistent with the effects of attention for the following reasons. If elevated attention was responsible for the increase in V1 responses in the high pre-stimulus response condition, we would have expected an improvement, not reduction, in perceptual accuracy (*Figure 3*). In addition, attention has been shown to reduce pairwise correlations, whereas we found an increase in correlations in the condition most likely to be associated with an increase in attention.

A recent study in area V4 of behaving monkey (*Engel et al., 2016*) involving laminar recordings using linear arrays reported striking Up and DOWN state transitions during wakefulness (both during passive fixation and attention). That is, neurons within a cortical column were highly synchronized throughout trials as they rapidly fired together for hundreds of ms, and then responded at a much slower rate (see also related studies in area V1 of anesthetized and fixating macaque [*Gutnisky et al., 2017*; *Arandia-Romero et al., 2016*]); the momentary phase of the local ensemble activity predicted behavioral performance in an attentional task. There are several major differences between the two studies: (i) the degree and extent of population synchrony reported by Engel et al. are far greater than those reported here, and UP and DOWN states were present throughout trials regardless of behavioral state. Such highly synchronized neuronal activity within a cortical column would likely increase correlated variability beyond previously reported correlation values in V1 and V4, and regardless of cortical layer (*Hansen et al., 2012*; *Nandy et al., 2017*; *Smith et al., 2013*). (ii) Engel et al. found that attention modulates the relative duration of UP/DOWN states without being involved in the emergence of those states. In contrast, the effects of cortical response state reported here are inconsistent with the changes in neural responses due to attention. (iii) Engel et al. reported that when neurons are in an elevated response state before and during stimulus presentation, behavioral performance in a change detection task is improved. While that result may be consistent with the modulation of neuronal responses by global states, such as arousal, we found an opposite relationship, that is, enhanced behavioral performance and sensory coding accuracy in the low response state.

An alternative interpretation of our results is that the pre-stimulus activity could be related to the working memory of the first stimulus (target). Indeed, it has been shown that working memory can parametrically represent a past stimulus and has rich dynamics during the delay period (*Harrison and Tong, 2009*; *Romo et al., 1999*). Thus, if the pre-stimulus activity is related to working memory it might covary with the orientation of the target stimulus. To rule out this confound we examined the relationship between the 200 ms pre-stimulus activity and the orientation of the target stimulus by decoding neural activity using a linear discriminant analysis. The inputs to the linear decoder were the firing rates of all the neurons in a session and, for each session, we ran 100 bootstrapping iterations. We used 70% of randomly selected samples for training and the remaining samples were used for validating. However, the average decoding performance of target orientation across sessions was $50.75 \pm 0.82\%$ ($p>0.1$, bootstrap test; best session performance was 55.90%, $p=0.1$). Furthermore, we repeated the same analysis by pooling all the neurons recorded across sessions – in this case the decoder performance was 50.39% ($p=0.48$). These results indicate that working memory performance during the delay period cannot explain the effects of cortical response state on sensory coding and behaviour.

Based on our decoder analysis in *Figure 5*, even when only one neuron is used to determine cortical state network performance is relatively high (around 82%). This high level of performance may

seem a bit surprising given that monkeys performed only around 60–70% for the most difficult orientation differences. However, decoder and perceptual performances cannot be directly compared quantitatively. Indeed, decoder performance represents the capacity of the neuronal population to discriminate stimulus orientation by correctly classifying the target and test stimuli as being different, not the capacity of the population to predict behavioral choices. This explains why decoder performance (quantifying the amount of orientation information encoded by the V1 population) is different, and higher, than behavioral performance (correct vs. incorrect responses). Importantly, the values of our decoding performance are consistent with previous V1 studies quantifying the amount of stimulus information encoded in population activity (e.g., *Poort and Roelfsema, 2009*). Furthermore, the mean choice probability of our individual neurons was 0.51 ± 0.04, which is in agreement with previous reports in V1 (*Nienborg and Cumming, 2006*). It can also be argued that although decoder performance is high, the state-dependent changes in neuronal performance (*Figure 5*) are smaller than the changes in behavioral performance (*Figure 3*). However, the fact that our state-dependent behavioral results are quantitatively stronger than the corresponding neural results may indicate that *knowledge* of the neurons' pre-stimulus activity (low vs. high) may be more determinant than the actual *information* encoded by relatively small groups of cells. This may either be due to the fact that we did not sample a large enough cell population, or we were unable to record the most discriminative neurons encoding the stimuli in the task.

One potential concern in our experiments is eye movements. For instance, larger fixational eye movements during the delay interval could possibly increase neuronal responses before stimulus presentation to account for the decrease in behavioral performance in the 'high' response state. The fixation pattern, including the speed and number of microsaccades, could also contribute to differences in neuronal responses and behavioral performance in the different cortical states. Our animals exhibited a small but significant deviation in eye position across trials (mean 0.07), which is very small relative to the size of the V1 receptive fields. Such small deviations in eye position are unlikely to cause significant changes in neuronal responses. However, to rule out the eye movement confound we performed the following analyses: (i) we examined whether pre-stimulus response states are associated with changes in the quality of fixation during the pre-stimulus (delay) interval, but found that the amplitude and velocity of eye movements were not statistically different in the low and high pre-stimulus states (p>0.1, paired t-test for both comparisons). (ii) we examined whether correct and incorrect behavioral responses are associated with different patterns of eye movements, but failed to detect a significant relationship between behavioral choice and frequency, amplitude, velocity, and variability of eye movements during the delay interval (both the mean standard deviation of eye position across trials and the standard deviation of the mean eye position in each trial) (p>0.1, Wilcoxon signed-rank test for each comparison). (iii) we calculated the correlation between the timing of the pre-stimulus response increase during the delay interval and the timing of eye movements, but found no significant relationship in any of our recorded sessions (r = 0.006 ± 0.004; p>0.1, Pearson correlation). Altogether, these analyses indicate that the cortical response state modulation of behavioral performance reported here cannot be attributed to saccadic eye movements.

If changes in arousal, as measured by pupil diameter, cannot explain the changes in pre-stimulus activity reported here, what type of mechanism could account for the observed trial fluctuations in V1 population response? One possibility is that cortical networks fluctuate spontaneously between different response states, and that state transitions represent an emergent property of local V1 circuits. In support of this mechanism, it has been suggested (*Harris and Thiele, 2011*) that a spontaneous, transient, increase in recurrent synaptic activity (*Sanchez-Vives and McCormick, 2000*; *Shu et al., 2003*) could cause an increase in cortical responses (UP state), and subsequent synaptic depression (*Chelaru and Dragoi, 2008*; *Crochet et al., 2005*) or after-hyperpolarizing potassium conductances (*Sanchez-Vives and McCormick, 2000*) could reduce network excitability (DOWN state). This mechanism is supported by computational models – for instance, it has been shown (*Holcman and Tsodyks, 2006*) that recurrent models relying on strong excitatory connections and depressing synapses can spontaneously transit between UP and DOWN states of different durations controlled by intrinsic local noise amplitude. Other local network mechanisms for the dynamics of cortical states have been proposed which rely on spike-timing-dependent plasticity (*Kang et al., 2008*) and integrate-and-fire models of excitatory and inhibitory neurons with nonlinear membrane currents (*Parga and Abbott, 2007*).

Although we measured the state of the population response based on the spiking activity of single neurons, another way to capture the fluctuations in network activity, ostensibly with less precision, is using local field potentials, LFPs (*Poulet and Petersen, 2008*; *Ecker et al., 2014*; *Goard and Dan, 2009*; *Kelly et al., 2010*). That is, previous studies have shown that the synchronized/desynchronized cortical state is associated with high/low LFP power in the low frequency range (0.5–10 Hz) and low/high power in the higher frequency range (10–30 Hz) (*Ecker et al., 2014*; *Vinck et al., 2015*; *Goard and Dan, 2009*). Thus, we examined the relationship between our trial-by-trial fluctuations in cortical response state and LFP power in each frequency band based on neural activity 0–200 ms before stimulus presentation. However, except for the theta band (4–8 Hz) which was associated with a small (<3%), but statistically significant change in LFP power (p=0.03, Wilcoxon rank-sum), we did not find statistically significant relationships between the two groups for delta (2–4 Hz), alpha (8–12 Hz), and beta (12–30 Hz) frequency bands (p>0.05, Wilcoxon rank-sum). Furthermore, we examined the relationship between the pre-stimulus response state and evoked LFP responses (*Figure 3—figure supplement 6*) – there was no significant difference in the evoked LFP total power during test stimulus presentation between the two groups (p=0.21, 2-way repeated measures ANOVA). Examining the LFP power in specific physiological frequency bands reveals only a small, statistically significant, difference in the upper portion of delta band (2–4 Hz, p=0.0006, Wilcoxon rank-sum), but no significant effects between the two groups in the higher frequency bands (p>0.05, Wilcoxon rank-sum).

Our results demonstrate that the response state of local cortical networks can be used to predict the coding of visual information and perceptual accuracy. This could lead to further experiments to causally manipulate the pre-stimulus cortical responses to impact the trial-by-trial population code and behavioral responses. Future research will elucidate whether the rapid fluctuations in population responses are coordinated across brain areas and whether the coordinated fluctuations are relevant for behavior. We propose that the relationship between fluctuations in cortical responses, network coding, and behavioral performance may be a component of a more general coding strategy in other sensory and downstream cortical areas. Indeed, given the similarities of the microcircuitry underlying different sensory modalities (*Douglas and Martin, 2004*; *Roe et al., 2015*), the control of network coding and behavior by pre-stimulus cortical activity could constitute a ubiquitous mechanism extending beyond vision.

## Materials and methods

### Behavioral experiments

All experiments in this manuscript were conducted in accordance with protocols approved by the National Institutes of Health and the Institutional Animal Care and Use Committee at The University of Texas Health Science Center at Houston. Three male monkeys (*Macaca mulatta*) were trained in a fixed delayed-match-to-sample task in which they had to indicate whether the orientation of two successively presented 4 deg circular sine-wave gratings had the same or different orientation. A fourth monkey was trained in the randomized delay experiment. In the fixed-delay task, after the monkeys maintained fixation for 100 ms, a target stimulus was flashed for 400 ms. The possible target orientations were 0, 45, 90 and 135°. During a delay of 1050 ms, the screen remained blank and monkeys were required to maintain fixation. In half of the trials, the test stimulus had the same orientation as the target ('match' condition). In the other half of the trials the target stimulus was randomly chosen within ±5° or ±10° of the target ('non-match' condition) and flashed for 200–400 ms. In the randomized delay task, the stimulus expectation effect is diminished (delay was randomized between 250–750 ms). We also increased the length of the fixation period to 400 ms to be able to compare the effects of the pre-target response state in the behavioral performance (*Figure 3E*).

All analyses were performed by separating trials into two groups, low and high pre-stimulus activity trials, depending on whether neuronal activity during the pre-stimulus period (200 ms) was lower or higher than the median pre-stimulus firing rate (this ensures that the number of trials in the two categories was equal). For each neuron, we determined the pre-stimulus activity and evoked response magnitude. In addition, we computed the change in performance on a session-by-session basis to eliminate potential bias. The number of possible combinations of subpopulation of size 1 to *n* for each session increases with the growth rate of a factorial function. As population size increases

(e.g., n = 13), sessions with larger numbers of cells would bias the results as they would weigh more into the pooled results. For this reason, we decided to average our results across sessions, and hence prevent the bias from individual sessions.

In all monkey experiments, eye position was continuously monitored using an infrared eye tracking system operating at 1 KHz (Eyelink Inc.). Monkeys fixated on a 0.1 deg dot in the center of screen throughout the trial (fixation instability larger than 0.2 deg caused the trial to abort). We examined whether pre-stimulus states are associated with changes in the quality of fixation (standard deviation of eye position, eye movement velocity, etc.) along the vertical and horizontal axes during the pre-stimulus interval, but found that eye movements were not statistically different in the low and high pre-stimulus states (p>0.1, paired t-test). We also examined whether correct and incorrect behavioral responses were associated with changes in eye movements, but failed to detect a significant relationship (p>0.1, Wilcoxon signed-rank test, by comparing the standard deviation of horizontal and vertical eye movements for correct and incorrect responses).

## Electrophysiological recordings

All experiments were conducted using a combination of in-house or Crist-grid electrode arrays (up to six electrodes) and laminar electrodes (Plextrode U-Probe, Plexon Inc) with 16 equally spaced contacts (100 μm inter-contact spacing). We recorded at cortical depths between 200 and 400 μm (monkey V1) with the electrode grid and from all depths with the linear electrode array. We recorded cells with orientation preferences spanning the entire orientation range (between 0–180°). Stimulus presentation was controlled by the Experimental Control Module (ECM, FHC Inc.). Neuronal and behavioral events were recorded using the Plexon system (Plexon Inc., Dallas, TX, USA).

Real-time neuronal signals were amplified recorded and stored with Multichannel Acquisition Processor system (MAP, Plexon Inc) at a sampling rate of 40 KHz and stored digitally. Units were identified by visual inspection in an oscilloscope and heard through a speaker. Waveforms that crossed a user-specified threshold (typically ~4 sd of the noise) were stored for further offline analyses. The spike waveforms were sorted using Plexon's offline sorter program (using waveform clustering based on parameters such as principle component analysis, spike amplitude, timing, width, valley and peak). *Figure 1—figure supplement 1* show examples of single units extracted in different experiments and the recording stability of units used in the behavioral experiments.

## Pearson correlation

The Pearson correlation $R(x,y)$ of two time series $x(n)$, $y(n)$, $n = 1,2,...,N$ is given by:

$$R(x,y) = \frac{\sum_{n=1}^{N} [x(n) - \bar{x}] (y(n) - \bar{y})}{\sigma_x \sigma_y}$$

where $\bar{x}$ and $\bar{y}$ are the means of $x$ and $y$, respectively, and $\sigma_x$ and $\sigma_y$ are the standard deviations of $x$ and $y$, respectively. We used the MATLAB function *corrcoef* to compute the Pearson correlation.

## Fisher linear discriminant analysis

Fisher linear discriminant (FLD) is a method to reduce the dimensionality of the data and to assess the neural classification performance in the low and high pre-stimulus conditions. We calculated the most discriminant dimension $w^*$ as:

$$w^* = S_w^{-1}(u_1 - u_2)$$

where $u_1$ and $u_2$ represent the mean vectors of spikes for all cells in the target and test condition, respectively. $S_w^{-1}$ represents the inverse of the total scatter matrix defined by:

$$S_w^{-1} = (S_1 + S_2)^{-1}$$

Where $S_1$ and $S_2$ represent the scatter matrices for target and test, respectively.

$$S_1 = (n-1) * cov(x_1)$$

$x_1$ and $x_2$ represent the matrices of spikes for the target and test stimuli, respectively. This analysis

allowed us to find the optimal line direction $w^*$ on which to project our data. We then plotted histograms in this dimension and fit a normal distribution using the *fitdist* function in MATLAB. To compute the 'difference in FLD means' in *Figure 5D*, we calculated the difference between the means in the fitted distributions. To compute the 'FLD pooled $\sigma$', we calculated the square root of the average variance of the fitted distributions.

The discriminability between two multivariate distributions using the FLD weight vector can be quantified with the multivariate generalization of $d^2$ (*Averbeck and Lee, 2006*; *Averbeck and Lee, 2004*; *Poor, 1994*) given by:

$$d^2 = \Delta\mu^T Q^{-1} \Delta\mu$$

Where $\Delta\mu$ is the vector difference in mean responses between the target and test orientation and Q is the pooled covariance matrix. Probability of correct classification ('With correlations') is given by:

$$erfc\left(-\sqrt{d^2}\right)/2$$

'Without correlations' represents the probability of correct classification while ignoring the effect of noise correlations using:

$$d^2_{shuffled} = \Delta\mu^T Q_d^{-1} \Delta\mu$$

where $Q_d$ is the diagonal covariance matrix obtained by setting the off-diagonal elements corresponding to correlations between neurons to 0. The quantity measures the information in a dataset of uncorrelated neural response and can be smaller or larger than $d^2$ (*Averbeck and Lee, 2006*).

## Statistics

Analysis methods are listed for each statistical comparison throughout the text. To assess the statistical significance of the probability that a given cell is in the same pre-stimulus state as the population, we computed the confidence interval (CI) of the binomial distribution:

$$CI_{low} = n * \left( 0.5 - \frac{1.96}{2} * \sqrt{\frac{1}{n}} \right)$$

$$CI_{high} = n * \left( 0.5 + \frac{1.96}{2} * \sqrt{\frac{1}{n}} \right)$$

Where n is the number of trials for each session.

## Acknowledgements

Research was supported by grants from National Eye Institute (NEI), ARRA funds from NEI, James S McDonnell Foundation, Pew Scholars Program (VD).

## Additional information

### Funding

| Funder | Grant reference number | Author |
| --- | --- | --- |
| National Institutes of Health | 1R01EY026156 | Valentin Dragoi |
| National Institute of Mental Health | 5R01MH086919 | Valentin Dragoi |
| James S. McDonnell Foundation | | Valentin Dragoi |
| Pew Charitable Trusts | Pew Scholars Program | Valentin Dragoi |

The funders had no role in study design, data collection and interpretation, or the decision to submit the work for publication.

## Author contributions
Diego A Gutnisky, Resources, Data curation, Software, Validation, Investigation, Writing—original draft; Charles Beaman, Data curation, Software, Validation, Investigation, Writing—original draft, Writing—review and editing; Sergio E Lew, Validation, Writing—review and editing; Valentin Dragoi, Conceptualization, Resources, Supervision, Funding acquisition, Methodology, Writing—original draft, Project administration, Writing—review and editing

## Author ORCIDs
Valentin Dragoi  http://orcid.org/0000-0002-9526-0926

## Ethics
Animal experimentation: This study was performed in strict accordance with the recommendations in the Guide for the Care and Use of Laboratory Animals of the National Institutes of Health. All of the animals were handled according to approved institutional animal care and use committee (IACUC) protocols of the University of Texas Health Science Center (protocol number AWC-17-0072). All surgery was performed under isofluorane anesthesia, and every effort was made to minimize suffering.

## Decision letter and Author response
Decision letter https://doi.org/10.7554/eLife.29226.020
Author response https://doi.org/10.7554/eLife.29226.021

# Additional files

## Supplementary files
• Transparent reporting form
DOI: https://doi.org/10.7554/eLife.29226.018

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
