## [Decision Letter]

Thank you for submitting your article "Cortical response states for enhanced sensory discrimination" for consideration by *eLife*. Your article has been reviewed by two peer reviewers, and the evaluation has been overseen by a Reviewing Editor and coordindated by myself as the Senior Editor. The following individual involved in review of your submission has agreed to reveal his identity: Alfonso Renart (Reviewer #2).

The reviewers have discussed the reviews with one another and the Reviewing Editor has drafted this letter to offer you an opportunity to respond to the serious issues identified by the reviewers. At this point, we ask that you write back with a detailed plan to address the essential points raised below and provide a time frame for the completion of these tasks. Your response will then be considered by the Board and the reviewers who will the issue a binding recommendation.

The reviewers found your research topic of interest for understanding the roles of variability of neural activity in sensory encoding and perception. That said, the reviewers and I found the way this was addressed was not especially novel: there was a general lack of depth in the analysis surrounding variability aspects. The reviewers and I would have liked to see a deeper connection between how variability shifted with high/low activity and the decoding performance. At this stage of your work, I am not sure of whether you would be able to address the reviewers' concerns. However, given the interest in the topic, we wish to learn how you could satisfy the reservations of the reviewers.

Essential revisions:

1) Baseline subtraction. Responses in the low pre-stimulus condition (LPSC) are stated to be smaller than in the HPSC. The rasters in Figure 1BC show a moderate effect of the stimulus in shaping the activity of the neurons, compared to spontaneous fluctuations. The higher rates in the HPSC can thus be due to higher activity in the baseline (by construction). It would be nice to know:a) What is the magnitude of baseline-corrected evoked responses.b) What is the relative magnitude of (baseline-subtracted) evoked responses compared to spontaneous fluctuations.

The decoding analysis in Figure 4 and Figure 5 could also be done with baseline-subtracted responses.

General point being that since activity changes quite a bit in the absence of the stimulus, a simple way to understand what is the responsibility of the stimulus over the evoked responses is to baseline subtract.

2) Dependence of the effect on difficulty. There are two things that shape performance: stimulus difficulty and, as shown by the authors, pre-stimulus activity (PSA). What is the relative contribution of each?

The authors provide very little information on this issue. In Figure 3 we see an example of a session with lower performance for 10° than for 5°. We assume this is not a representative session. And then when combining across sessions, the overall performance is normalised away. It would be useful to know just performance depends on difficulty.

The effect of PSA is presumably added on top of this. But does its magnitude change for the two difficulties. We currently can't tell because we only see normalised performance. For this quantity, the effect of PSA is smaller for 10°. Is it really smaller, or is it only smaller relative to the increased performance for easier trials?

3) Decoding accuracy vs performance. In Figure 5, we see that performance of the decoder, even with only 1 neuron (I'm guessing averaged across sessions/monkeys?) is ~ 0.82. That's quite high. Again, the limited info we are given is that the monkey performs at ~ 70%? Is this really true that a single neuron does consistency better than the animal?

4) Shuffling the data (removing correlations) leads to an overall decrease in performance. This is interesting, but not really explored. It suggests that the neurons that authors are simultaneously recording have typically negative signal correlations. Is this really true? This is surprising because in primate V1, one might assume that nearby cells would have similar tuning curves, which, in the presence of net positive correlations (Figure 5) would presumably have led to a situation where correlations were detrimental. Can the authors clarify what's going on?

5) LFP analysis. Seems puzzling to look at power in different bands on a phenomenon that appears intrinsically transient (brief evoked responses). Can the authors just look at changes in evoked LFP? Both PSA and evoked responses can be evaluated at the level of the LFP. It would be interesting to do so. Furthermore, it appears to be the only 'global' (or at least more global than the single spikes) signal that the authors record, so it can help clarify some of their claims (see below) on the spontaneous fluctuations being locally generated. Related to this, concerning the Crist-grid arrays, how far are the tips? Can we learn something about the local/global nature of the signal by comparing LFP across electrodes?

6) Local/Global fluctuations. Several places (Discussion section), the authors make a contrast between the type of fluctuations that they analyse (which they say are local, and even that they originate in the local circuit!!) and 'global' up/down fluctuations. But what is the evidence behind these local/global claims? One would need to record a 'global signal' (either a local ePhys signal at different distant locations, or wide-field imaging) and show that it is weakly correlated with a particular local signal to make these claims. Do we have any information about the global state of the cortex, or at least the visual cortex? How do we know the global (in the sense of being correlated across the recorded neurons) signal the authors record and analyse, is not correlated with a similar signal a few mm away? Unless some evidence is provided, the statements about local/global nature and origin of the signal analysed seem unfounded.

7) Last paragraph of the Discussion section: The authors write: "Altogether, our results demonstrate that the variability in pre-stimulus cortical activity is not simply noise but has a dynamic structure that controls how incoming sensory information is optimally integrated with ongoing processes to guide network coding and behavior." This statement is puzzling. What does it mean it is not simply noise? A starting point would be to show that it cannot be predicted by anything, but the authors do not do any analysis of this type. Maybe they mean that the fluctuations are not uncorrelated across cells? The authors should either be more accurate/explicit, or remove this sentence. It's a not-so-good ending for a nice study!)

8) The changes in the probability of correct detection (PCD, Figure 5) are very modest, especially compared to the actual conditioned (high vs low) performance of the animal in the detection task (Figure 3). This is sort of a letdown, perhaps suggesting that FLD is insufficient to capture the main effects, or that simply the number of neurons are too small. In any event, while the effects are certainly statistically significant the take home message from Figure 5 is not overly impressive.

9) The noise correlation vs coding results are puzzling. The noise correlations are higher in the high state than the low state (Figure 5), and the PCD is higher in the low state than the high state (Figure 5, left). This seems ok However, when the spike trains are trial shuffled, artificially removing the noise correlations, then PCD (as computed from the Fisher linear estimate) actually decreases. Thus, the excess noise correlations in the high state are deleterious to coding (Figure 5, left, red vs. blue), yet the complete absence of correlations is also deleterious to coding (Figure 5, blue, right vs left). This suggests that either how the structure of the covariance changes is very important, and shuffling trials is very different than the decorrelation from high to low, or the shift in response gain is more impressive than any correlation change.

The authors are completely aware of this (subsection “Cortical state influences encoded information”) and they conclude that the main effect is that the low state has a higher gain/sensitivity (Figure 5). That is fine, except then the whole paper now loses steam. The higher sensitivity to punctate inputs in the low state compared to the high state has been shown in the somatosensory system (Fanselow and Nicolelis 1999; Sachdev, Ebner, and Wilson, 2004). Further, the authors recently published a related manuscript in cerebral cortex (2016) that dissected coding in terms of up and down states, and gain changes were critical there also. But those results were in cat and did not consider animal behavior during a task. The current manuscript may be the first demonstration of this in awake primate V1.

In the end, while the correlation and variability analysis is important for the evaluation of the FLD, the state dependent changes in variability seem less important for the ultimate shift in discrimination.

[Editors' note: further revisions were requested prior to acceptance, as described below.]

Thank you for sending your article entitled "Cortical response states for enhanced sensory discrimination" for peer review at *eLife*. Your article is being evaluated by two peer reviewers, and the evaluation is being overseen by a Reviewing Editor and Richard Ivry as the Senior Editor.

The reviewers had a favorable response to the revision. There were a few, relatively minor raised by one of the reviewers that I think should be addressed.

1) There are no labels in the Supplementary figures which allows one to identify them

2) Although it is a good idea to do the pupil analysis, I don't think that showing the (lack of) relationship between the pipil diameter and the pre-stim baseline is good enough evidence that the changes pre-stim activity are local. Pre-stim baseline and pupil diameter are two different signals with presumably different time-constants etc. I think it is fair to conclude that changes in arousal as indexed by pupil diameter do not seem to explain the changes in pre-stim baseline. However, in order to conclude that the pre-stim signal is local one would have to measure it at distant cortical locations and show that as distance grows, the correlation between the different pre-stim baselines decreases. I would thus suggest that the authors re-phrase the last sentence of subsection “Pooling neurons improves the predictability of behavioral responses” and first sentence of the eleventh paragragh of the Discussion section. Sorry for being picky but the local-global nature of this signals is quite important and I don't think sloppy statements in this regards have room here.

3) Discussion section. I don't think the authors can say this, as attention in their task is uncontrolled.

4) Results section. I may be missing something, but in my opinion what Figure 1—figure supplement 4 shows is that the baseline subtracted evoked response is larger with low pre-stim activity than with high pre-stim activity, NOT that stim-driven fluctuations are smaller than evoked fluctuations (and in any event this last finding would not imply non-linearity…).

---

## [Author Response]

Essential revisions:1) Baseline subtraction. Responses in the low pre-stimulus condition (LPSC) are stated to be smaller than in the HPSC. The rasters in Figure 1BC show a moderate effect of the stimulus in shaping the activity of the neurons, compared to spontaneous fluctuations. The higher rates in the HPSC can thus be due to higher activity in the baseline (by construction). It would be nice to know:a) What is the magnitude of baseline-corrected evoked responses.b) What is the relative magnitude of (baseline-subtracted) evoked responses compared to spontaneous fluctuations.The decoding analysis in Figure 4 and Figure 5 could also be done with baseline-subtracted responses.General point being that since activity changes quite a bit in the absence of the stimulus, a simple way to understand what is the responsibility of the stimulus over the evoked responses is to baseline subtract.

We agree that the difference between evoked responses in the high vs. low pre-stimulus condition is moderate, although highly statistically significant (P < 0.0001, paired t-test). We also agree that, as expected, the higher evoked responses in the high pre-stimulus condition is due to the higher baseline activity. We followed thereviewers’ advice and subtracted pre-stimulus activity from evoked responses. When we compared the baseline-subtracted evoked responses in the two pre-stimulus states, we found that stimulus-driven fluctuations in neuronal responses are smaller than the fluctuations in ongoing activity (high vs. low pre-stimulus response, Figure 1—figure supplement 4). This argues that, during wakefulness, evoked responses cannot be predicted from the linear summation of the deterministic response and the preceding ongoing activity, as suggested by earlier studies in anesthetized animals. This is now mentioned in the Introduction and Figure 1—figure supplement 4.

Regarding the reviewers’ suggestion to decode the population activity by baseline-subtracting the stimulus-evoked-responses, we have performed this control analysis. However, we were unsure about the relevance of such an analysis. Our point is that in brain networks, a downstream decoder of the population response has full access to the evoked activity. Pre-subtracting ongoing activity (our response state variable) from the evoked response is a complex operation that downstream neurons would need to perform, possibly relying on complex mechanisms (unknown to us). A more parsimonious approach is that downstream neurons read out the stimulus-evoked responses without performing a baseline subtraction. Importantly, one of our stated goals in the manuscript is to understand the impact of pre-stimulus state on the network coding of sensory information – by subtracting ongoing activity from the stimulus-evoked responses we would fail to address the very issue we seek to investigate. Nonetheless, we followed the reviewers’ suggestion and retrained the neural decoder using the baseline-subtracted evoked responses. As expected, we found that the difference between decoder performances in the low vs. high pre-stimulus states was abolished when pre-stimulus activity was subtracted from the evoked response (Figure 5—figure supplement 1). This result indicates that removing the influence of ongoing activity on evoked responses renders the decoder of population activity insensitive to changes in pre-stimulus cortical state. The accompanying text in the subsection “Cortical state influences encoded information” describes this analysis (without discussing the logistical points made above).

2) Dependence of the effect on difficulty. There are two things that shape performance: stimulus difficulty and, as shown by the authors, pre-stimulus activity (PSA). What is the relative contribution of each?The authors provide very little information on this issue. In Figure 3 we see an example of a session with lower performance for 10° than for 5°. We assume this is not a representative session. And then when combining across sessions, the overall performance is normalised away. It would be useful to know just performance depends on difficulty.The effect of PSA is presumably added on top of this. But does its magnitude change for the two difficulties. We currently can't tell because we only see normalised performance. For this quantity, the effect of PSA is smaller for 10°. Is it really smaller, or is it only smaller relative to the increased performance for easier trials?

The reviewers are correct: both the task difficulty and cortical response state influence behavioral performance. How are these two variables expected to interact? When the task difficulty is high (e.g., 5° or 10° orientation discrimination), we expect a high impact of cortical state on performance (because of the large number of incorrect responses). In contrast, when the task difficulty is low (e.g., 20° orientation discrimination), the relative number of ‘incorrect’ response trials is very low, and the impact of cortical state would be impossible to determine. We now provide more information about behavioral performance as a function of task difficulty – as expected, animals performed more poorly (Figure 3—figure supplement 3) when they discriminated small orientation differences ( ± 5o; mean across sessions: 64.58 ± 2.19% correct responses) compared to less difficult task conditions ( ± 10°; mean across sessions: 79.17 ± 1.85% correct responses, P = 4.03 10-5, ranksum test). The impact of pre-stimulus response state on behavioral performance did depend on task difficulty. As shown in Figure 3, the change in behavioral performance (low vs. high pre-stimulus state) is significantly larger for the more difficult orientation difference (P < 0.01, Wilcoxon signed-rank). This is now clarified in the subsection “Pooling neurons improves the predictability of behavioral responses” and accompanying Figure 3—figure supplement 3.

3) Decoding accuracy vs performance. In Figure 5, we see that performance of the decoder, even with only 1 neuron (I'm guessing averaged across sessions/monkeys?) is ~ 0.82. That's quite high. Again, the limited info we are given is that the monkey performs at ~ 70%? Is this really true that a single neuron does consistency better than the animal?

We acknowledge that the reviewers might have been confused by Figure 5. There are several reasons that could have contributed for this confusion, which is now clarified in the revised manuscript:

i) The text accompanying Figure 5 was not clearly written – what is represented is the mean performance of a decoder using all the neurons in the population, for which cortical state was measured based on the prestimulus activity of a variable number of cells in each session (from 1 to 13). For instance, decoder performance corresponding to n=1 represents the mean population decoder performance, considering all the cells in the population, in which the low and high pre-stimulus activity trials were selected based on the prestimulus response of only one cell (subsequently averaging the decoder performances corresponding to low vs. high pre-stimulus state for each neuron in the population). Since Figure 2 already shows that the state of one cell is coupled to local population activity, the decoder performance is relatively high (around 0.82). This is clarified in the subsection “Pooling neurons improves the predictability of behavioral responses”.

ii) Decoder performance represents the capacity of the neuronal population to discriminate stimulus orientation (by correctly classifying the target and test stimuli as being different), not the capacity of the population to predict behavioral choices. This explains why decoder performance (quantifying the amount of orientation information encoded by the V1 population) is different (and higher) than behavioural performance (correct vs. incorrect responses). Importantly, the values of our decoding performance are consistent with previous studies that have quantified the amount of stimulus information encoded in population activity. For instance, in a study performed in V1 that measured the neural population discrimination (or classification performance) of simultaneously recorded neurons (using similar methodology), Poort and Roelfsema (2009) reported population discriminability values similar to our reported values. That is, they found *d^2^* values of 13 when including up to 6 recording sites, which corresponds to a classification rate of 96% (even higher than that reported in our study). This is now discussed in the Discussion section.

iii) A more direct comparison between behavioral performance and the performance of single neurons is given by the correlation between single neuron response fluctuations and behavioral choices, typically termed choice probability (CP). This measure has been amply analyzed in previous studies in a variety of cortical areas, and CPs in V1 have been found to be close to chance level (~0.51, e.g., Nienborg and Cumming, 2006). To confirm that our data is in agreement with these previous reports, we computed the neural choice probability (CP) in our data set for single neurons and found that the mean CP of our cells is 0.51 ± 0.04, which is in direct agreement with previously published literature. This is discussed in the Discussion section.

4) Shuffling the data (removing correlations) leads to an overall decrease in performance. This is interesting, but not really explored. It suggests that the neurons that authors are simultaneously recording have typically negative signal correlations. Is this really true? This is surprising because in primate V1, one might assume that nearby cells would have similar tuning curves, which, in the presence of net positive correlations (Figure 5) would presumably have led to a situation where correlations were detrimental. Can the authors clarify what's going on?

The reviewer is correct that it is somewhat surprising that removing correlations reduces the overall probability of correct classifications. However, there are significant negative noise correlations in our population of cells (30% of our correlation coefficients are negative, Figure 5). This is not unusual since neurons were recorded within 2 mm of cortex (correlations would be typically expected to be positive within 200-500 um cortical distance). Also, there is no reason to assume that cells within 2 mm of cortex will have positive signal correlations, i.e., similar tuning (we found that 37% of cell pairs have negative signal correlations). It is unclear whether removing a mix of negative and positive correlations would automatically increase, rather than decrease, network performance. Although noise correlations are generally believed to be detrimental for coding, this is typically true when correlations are high and positive. Removing negative correlations has already been shown to decrease the amount of information encoded in population activity (Chelaru and Dragoi, 2017; Rosenbaum et al., 2017). Furthermore, correlations can increase or decrease the amount of encoded information depending on the structure of correlations across a population of cells, not the absolute magnitude of correlation coefficients (Moreno-Bote et al., 2014). However, despite the fact that removing correlations decreases encoded information, the difference in the probability of classification between the two response states remained relatively unchanged (Figure 5), although the difference in classification performance between the low and high pre-stimulus states remained significant. This is now clarified in the subsection “Cortical state influences encoded information”.

5) LFP analysis. Seems puzzling to look at power in different bands on a phenomenon that appears intrinsically transient (brief evoked responses). Can the authors just look at changes in evoked LFP? Both PSA and evoked responses can be evaluated at the level of the LFP. It would be interesting to do so. Furthermore, it appears to be the only 'global' (or at least more global than the single spikes) signal that the authors record, so it can help clarify some of their claims (see below) on the spontaneous fluctuations being locally generated. Related to this, concerning the Crist-grid arrays, how far are the tips? Can we learn something about the local/global nature of the signal by comparing LFP across electrodes?

We followed the reviewers’ advice and examined the relationship between the pre-stimulus response state and evoked LFPs, asking whether the high or low pre-stimulus activity impacts stimulus-evoked LFP responses. However, we did not find significant differences in total power of evoked LFPs during test stimulus presentation (p=0.21, 2-way repeated measures ANOVA). We next examined whether the LFP power in specific physiological frequency bands is influenced by pre-stimulus activity, but only found a small, significant, difference in the upper portion of delta band (2-4 Hz). These results are described in more detail on page 20, last paragraph, and Figure 3—figure supplement 6. For the Crist grid array recordings, the electrode tips were within 2 mm of each other. We agree with the reviewers that the LFP signals cannot be substituted for global signals. Regarding the question about local/global nature of our neural signals, we analyzed the impact of general arousal (a global signal) on pre-stimulus activity (see our response below to point 6).

6) Local/Global fluctuations. Several places (Discussion section), the authors make a contrast between the type of fluctuations that they analyse (which they say are local, and even that they originate in the local circuit!!) and 'global' up/down fluctuations. But what is the evidence behind these local/global claims? One would need to record a 'global signal' (either a local ePhys signal at different distant locations, or wide-field imaging) and show that it is weakly correlated with a particular local signal to make these claims. Do we have any information about the global state of the cortex, or at least the visual cortex? How do we know the global (in the sense of being correlated across the recorded neurons) signal the authors record and analyse, is not correlated with a similar signal a few mm away? Unless some evidence is provided, the statements about local/global nature and origin of the signal analysed seem unfounded.

We agree with the reviewers that our references to local vs. global fluctuations in the original manuscript were not warranted. We have now directly addressed this issue by examining whether the fluctuations in cortical response state are correlated with global arousal, or they reflect local changes in network activity. Therefore, we performed additional analyses in which we examined the correlation between pre-stimulus response to the test stimulus and pupil size, which is a measure of global arousal. That is, pupil dilation is associated with increased arousal, whereas pupil constriction is associated with diminished arousal. In a subset of sessions (n=11), examining the 200-ms pre-stimulus test response reveals that out of 131 cells, only 12 exhibited significant trial-by-trial correlation with mean pupil size (P < 0.05, Pearson correlation; of these cells, 7 showed a positive correlation between the pre-stimulus response and pupil size). However, when we calculated the population response, none of the sessions exhibited a significant trial correlation between the population pre-stimulus response and pupil size (P > 0.05, Pearson correlation). Additionally, when we divided trials into two groups based on median pupil size (low vs. high pupil size), there was no significant difference between the mean pre-stimulus firing rates in the two groups (P > 0.1, Wilcoxon signed rank test, pooling all the cells in our sample). This analysis demonstrates that global fluctuations in brain state, measured by pupil size, is not related to the trial-by-trial fluctuations in pre-stimulus response state discussed here. Thus, trial fluctuations in cortical response state likely have a local circuit origin rather than reflecting global cortical states controlled by arousal. This is addressed in the subsection “Pooling neurons improves the predictability of behavioral responses”.

7) Last paragraph of the Discussion section: The authors write: "Altogether, our results demonstrate that the variability in pre-stimulus cortical activity is not simply noise but has a dynamic structure that controls how incoming sensory information is optimally integrated with ongoing processes to guide network coding and behavior." This statement is puzzling. What does it mean it is not simply noise? A starting point would be to show that it cannot be predicted by anything, but the authors do not do any analysis of this type. Maybe they mean that the fluctuations are not uncorrelated across cells? The authors should either be more accurate/explicit, or remove this sentence. It's a not-so-good ending for a nice study!)

We agree with the reviewers that stating that the response fluctuations we record are not simply noise may be inadequate. We have now completely rewritten the last paragraph of the Discussion section to remediate this error.

8) The changes in the probability of correct detection (PCD, Figure 5) are very modest, especially compared to the actual conditioned (high vs low) performance of the animal in the detection task (Figure 3). This is sort of a letdown, perhaps suggesting that FLD is insufficient to capture the main effects, or that simply the number of neurons are too small. In any event, while the effects are certainly statistically significant the take home message from Figure. 5 is not overly impressive.

It is true that the effects in Figure 3 are somewhat more impressive than those in Figure 5. We could also argue that Figure 5 inset may be more impressive than Figure 5.e., there is a clearly significant difference between the capacity of large populations of cells to exhibit state-dependent changes in encoded information in comparison with small populations. In any case, the fact that our behavioral results are quantitatively stronger than the neural results could indicate that knowledge of the neurons’ pre-stimulus activity may be more determinant than the actual information encoded by small groups of V1 cells. This may be because, as the reviewer suggests, we do not sample a large enough cell population, or we were unable to record the most discriminative neurons. This issue is now discussed in the Discussion section.

9) The noise correlation vs coding results are puzzling. The noise correlations are higher in the high state than the low state (Figure 5), and the PCD is higher in the low state than the high state (Figure 5, left). This seems ok However, when the spike trains are trial shuffled, artificially removing the noise correlations, then PCD (as computed from the Fisher linear estimate) actually decreases. Thus, the excess noise correlations in the high state are deleterious to coding (Figure 5, left, red vs. blue), yet the complete absence of correlations is also deleterious to coding (Figure 5, blue, right vs left). This suggests that either how the structure of the covariance changes is very important, and shuffling trials is very different than the decorrelation from high to low, or the shift in response gain is more impressive than any correlation change. The authors are completely aware of this (subsection “Cortical state influences encoded information”) and they conclude that the main effect is that the low state has a higher gain/sensitivity (Figure 5).

This point is similar to point 4 above (which we have already discussed).

That is fine, except then the whole paper now loses steam. The higher sensitivity to punctate inputs in the low state compared to the high state has been shown in the somatosensory system (Fanselow and Nicolelis 1999; Sachdev, Ebner, and Wilson, 2004). Further, the authors recently published a related manuscript in cerebral cortex (2016) that dissected coding in terms of up and down states, and gain changes were critical there also. But those results were in cat and did not consider animal behavior during a task. The current manuscript may be the first demonstration of this in awake primate V1.In the end, while the correlation and variability analysis is important for the evaluation of the FLD, the state dependent changes in variability seem less important for the ultimate shift in discrimination.

We have now included the references suggested by the reviewer. However, although the two papers that were mentioned are certainly interesting, they make different points that those in our manuscript. The first study (by Fanselow and Nicolelis) was performed in rat S1 and the conclusion was that behavioral state (quiet immobility or whisker movement) renders the somatosensory system optimally tuned to detect the presence of single vs. rapid sequences of tactile stimuli. The second study (by Sachdev et al.), also performed in rat S1, concluded that postsynaptic potentials are stronger in the hyperpolarized “down” state (contrary to our findings that evoked responses are stronger when pre-stimulus response is in a “high” state). This is now discussed in the Discussion section. Importantly, none of the papers mentioned by the reviewer, including the Gutnisky et al., (2016) study, have analyzed how population activity relates to the amount of encoded information and to behavioral performance. In addition, our study examines the coupling between individual neurons and population activity and the relationship with behavior (Figure 2, which leads to all subsequent analyses), which was never attempted in other studies of cortical or subcortical function.

[Editors' note: further revisions were requested prior to acceptance, as described below.]

1) There are no labels in the Supplementary figures which allows one to identify them

We have added labels to the Supplementary Figures.

2) Although it is a good idea to do the pupil analysis, I don't think that showing the (lack of) relationship between the pipil diameter and the pre-stim baseline is good enough evidence that the changes pre-stim activity are local. Pre-stim baseline and pupil diameter are two different signals with presumably different time-constants etc. I think it is fair to conclude that changes in arousal as indexed by pupil diameter do not seem to explain the changes in pre-stim baseline. However, in order to conclude that the pre-stim signal is local one would have to measure it at distant cortical locations and show that as distance grows, the correlation between the different pre-stim baselines decreases. I would thus suggest that the authors re-phrase the last sentence of subsection “Pooling neurons improves the predictability of behavioral responses” and first sentence of the eleventh paragraph of the Discussion section. Sorry for being picky but the local-global nature of this signals is quite important and I don't think sloppy statements in this regards have room here.

The reviewer is correct – we have rephrased our claims in the subsection “Pooling neurons improves the predictability of behavioral responses” and in the first sentence of the eleventh paragraph of the Discussion, as suggested in the review. We now write “This analysis demonstrates that global fluctuations in brain state, measured by pupil size, do not seem to explain the trial-by-trial fluctuations in pre-stimulus response state discussed here” and “If changes in arousal, as measured by pupil diameter, cannot explain the changes in pre-stimulus activity reported here, what type of mechanism could account for the observed trial fluctuations in V1 population response? One possibility is that cortical networks fluctuate spontaneously between different response states, and that state transitions represent an emergent property of local V1 circuits…”.

3) Discussion section. I don't think the authors can say this, as attention in their task is uncontrolled.

While attention is not controlled (i.e., there is no ‘unattended’ condition as animals are always engaged in the task), we did consider attention as a possible variable responsible for our results. As written in the Discussion section, “one could possibly link the trial-by-trial fluctuations in cortical responses examined here to trial fluctuations in attention. Indeed, attention has been shown to modulate neuronal responses and sensitivity as well as response variability and pairwise correlations in early and mid-level visual cortical areas. However, our results are inconsistent with the effects of attention for the following reasons. If elevated attention was responsible for the increase in V1 responses in the high pre-stimulus response condition, we would have expected an improvement, not reduction, in perceptual accuracy. In addition, attention has been shown to reduce pairwise correlations, whereas we found an increase in correlations in the condition most likely to be associated with an increase in attention”. We have rephrased the statement in the Discussion, mentioned by the reviewer, writing that “…the effects of cortical response state reported here are inconsistent with the changes in neural responses due to attention.”.

4) Results section. I may be missing something, but in my opinion what Figure 1—figure supplement 4 shows is that the baseline subtracted evoked response is larger with low pre-stim activity than with high pre-stim activity, NOT that stim-driven fluctuations are smaller than evoked fluctuations (and in any event this last finding would not imply non-linearity…).

We apologize for the confusion. While the reviewer is correct that Figure 1—figure supplement 4 shows that “the baseline subtracted evoked response is larger with low pre-stim activity than with high pre-stim activity”, we are correct too by stating that the suppl. figure also shows that the stimulus-driven fluctuations are smaller than evoked fluctuations. Indeed, as described in the caption for Figure 1—figure supplement 4, the scatter plot shows that evoked_low – baseline_low > evoked_high – baseline_high.(as the reviewer correctly points out). However, this inequality is equivalent to evoked_high – evoked_low < baseline_high – baseline_low. That is, fluctuations in evoked responses during wakefulness (evoked_high – evoked_low) are smaller than the fluctuations in pre-stimulus activity (baseline_high – baseline_low). The reviewer is correct that this does not imply nonlinearity. Therefore, we have corrected our sentence in the Results section (to which the reviewer refers to) by writing that “…evoked responses cannot be predicted from the simple addition of the deterministic response and the preceding ongoing activity” (replacing the previous “linear summation” wording, which might have implied nonlinearity).